# FAIR OFF-POLICY LEARNING FROM OBSERVATIONAL DATA

## ABSTRACT

Algorithmic decision-making in practice must be fair for legal, ethical, and societal reasons. To achieve this, prior research has contributed various approaches that ensure fairness in machine learning predictions, while comparatively little effort has focused on fairness in decision-making, specifically off-policy learning. In this paper, we propose a novel framework for *fair off-policy learning*: we learn decision rules from observational data under different notions of fairness, where we explicitly assume that observational data were collected under a different – potentially discriminatory – behavioral policy. For this, we first formalize different fairness notions for off-policy learning. We then propose a neural network-based framework to learn optimal policies under different fairness notions. We further provide theoretical guarantees in the form of generalization bounds for the finite-sample version of our framework. We demonstrate the effectiveness of our framework through extensive numerical experiments using both simulated and real-world data. Altogether, our work enables algorithmic decision-making in a wide array of practical applications where fairness must be ensured.

## 1 INTRODUCTION

Algorithmic decision-making in practice must avoid discrimination and thus be fair to meet legal, ethical, and societal demands (Nkonde, 2019; De-Arteaga et al., 2022; Corbett-Davies et al., 2023). For example, in the U.S., the Fair Housing Act and Equal Credit Opportunity Act stipulate that decisions must not be subject to systematic discrimination by gender, race, or other attributes deemed as sensitive.

However, research from different areas has provided repeated evidence that algorithmic decision-making is often *not* fair. A prominent example is Amazon's tool for automatically screening job applicants that was used between 2014 and 2017 (Dastin, 2018). It was later discovered that the underlying algorithm generated decisions that were subject to systematic discrimination against women and thus resulted in a ceteris paribus lower probability of women being hired.

Ensuring fairness in off-policy learning is subject to inherent challenges. The reason is that off-policy learning is based on *observational data that observational data may ingrain existing bias from historical decision-making.*[1] Hence, one challenge is that the resulting policy must be fair despite the observational data being collected under a different – potentially discriminatory – behavioral policy. Furthermore, one may erroneously think that a naïve approach to achieving fairness in algorithmic decision-making is to simply omit the sensitive attribute from the observational data. For instance, to avoid bias against women, one would prevent off-policy learning from having access to a variable that stores the gender of an individual. However, in observational data, other variables may act as proxies for gender, and, hence, the learned policy may still lead to discrimination due to the underlying data-generating process (Kilbertus et al., 2017). Hence, a custom approach for handling sensitive attributes in off-policy learning is needed.

In this paper, we propose a novel framework for *fair off-policy learning from observational data*. Specifically, we learn fair decision rules from observational data where the observational may be

---

[1]The term "bias" can have different meanings. Here, we use bias can refer to algorithmic bias, where algorithms discriminate against individuals from certain sensitive groups. This is in contrast to the statistical bias of estimators, e.g., due to confounded data.

collected under a different – potentially discriminatory – behavioral policy. To the best of our knowledge, ours is the first neural approach to fair off-policy learning. In our off-policy learning framework, fairness may enter at two different stages, namely with respect to both the action and the policy value.

1. *Fairness with respect to the action* ensures that individuals with different sensitive attributes but otherwise equal characteristics receive the same decision. In other words, the choice of action is independent of the sensitive attribute. For example, in credit lending, this means that a woman and a man, each with the same academic subject, have the same chance that their student loan is approved. We later refer to this notion as "action fairness".

2. *Fairness with respect to the policy value* allows us to express fairness in the way that we consider the utility (i.e., the policy value) for each sensitive group. Hence, individuals with different sensitive attributes achieve, on average, a similar utility. For example, this may allow governments to account for the fact that some sub-populations have been historically under-represented. Hence, as women have a lower propensity than men to pursue academic careers in subjects related to technology, governments may want to strategically incentivize women through student loans so that the long-term benefit for society is maximized. We refer to this notion as "value fairness". Later, we introduce two variants of value fairness that build upon envy-free fairness and max-min fairness.

Our **contributions**[2] are three-fold. (1) We first introduce and formalize different fairness notions tailored to our setting, namely fair off-policy learning from observational data. We also provide a theoretical understanding of how these fairness notions interact in our setting. (2) We then propose a neural framework, called FairPol, to learn optimal policies under these fairness notions. Specifically, we leverage fair representation learning in combination with custom training objectives so that the resulting policies satisfy our fairness notions. (3) We provide theoretical learning guarantees in the form of generalizations bounds for FairPol. We also evaluate the effectiveness of our framework through extensive numerical experiments using both simulated and real-world data.

## 2 RELATED WORK

We provide an overview on related work on off-policy learning from observational data, both in the standard machine learning and algorithmic fairness literature. For further background on algorithmic fairness and fairness in utility-based decision models (e.g., reinforcement learning), we refer to Appendix A.

**Off-policy learning:** Off-policy learning typically aims to determine optimal policies from observational data by maximizing the so-called policy value (e.g., Kallus, 2018; Athey & Wager, 2021). The policy value is a causal quantity, which can be identified from observational data under certain assumptions (see Section 3). There are three standard methods for estimating the policy value: (1) The direct method (DM) (Qian & Murphy, 2011; Bennett & Kallus, 2020); (2) The inverse propensity score weighted (IPW) method (Kallus, 2018); and (3) The doubly robust (DR) (Athey & Wager, 2021; Chernozhukov et al., 2022). Several works propose extensions of the three standard methods for specific settings, such as unobserved confounding (Kallus & Zhou, 2018a; Bennett & Kallus, 2019) or distribution shifts (Hatt et al., 2022; Kallus et al., 2022), or overlap violations (Kallus, 2021). Different from our work, none of the above works deals with algorithmic fairness in off-policy learning.

**Fair representation learning:** A popular approach to achieve fairness in machine learning models is to remove the algorithmic bias incorporated in the training data by producing a new, fair representation of the data (e.g., Zemel et al., 2013; Locatello et al., 2019). For this, one typically uses neural networks that learn such a fair representation, and, then, the fair representation is used as input to the actual prediction model. For instance, statistical parity can be achieved by producing a new representation of the data that is non-predictive of the sensitive attributes using probabilistic models (Creager et al., 2019) or adversarial learning methods (Madras et al., 2018). In our work, we adapt fair representation to satisfy parts of our fairness constraints. However, our main contribution is *not* a new method for fair representation learning, but rather we adapt fairness notions and provide an understanding of how these fairness notions interact in the context of off-policy learning.

---

[2]Code is available at `https://anonymous.4open.science/r/FairPol-402C`.

**Algorithmic fairness for off-policy learning from observational data:** Closest to our framework is the work by Viviano & Bradic (2023), which studies fair off-policy learning for Pareto-optimal policies. There are two major differences to our work: (1) Viviano & Bradic (2023) propose to maximize fairness over the set of Pareto-optimal policies. Here, Pareto optimality is defined so that the policy value of one sensitive group cannot be improved without reducing the policy value for the opposite group. In contrast, we propose to incorporate our fairness notions by adjusting the off-policy learning objective (value fairness), and then maximize this objective over the class of so-called action fair policies. (2). The approach from Viviano & Bradic (2023) is restricted to learning *linear* policies, while our framework enables learning arbitrarily *non-linear* policies. This is possible because we can incorporate the action fairness constraint by leveraging fair representation learning to obtain a representation independent of the sensitive attribute, which can be used in a second step to train an optimal policy.

## 3   PROBLEM SETTING

We build upon the standard setting for policy learning from observational data (e.g., Kallus, 2018; Athey & Wager, 2021). We consider observational data $(X_i, S_i, A_i, Y_i)_{i=1}^n$ sampled i.i.d. from a data-generating process $(X, S, A, Y) \sim \mathbb{P}$, which consists of user-specific covariates $X \in \mathcal{X} \subseteq \mathbb{R}^p$, sensitive attributes $S \in \mathcal{S}$, an action $A \in \{0, 1\}$, and an outcome $Y \in \mathbb{R}$.[3] For example, in credit lending, one could model the credit score of an applicant by $X$, the gender or age as a sensitive attribute $S$, a decision $A$ whether to approve or reject the loan, and a profit $Y$ for the lending institution. The causal graph from our setting is shown in Fig. 1. Note that modeling the action $A$ as a binary variable is consistent with previous literature (e.g., Kallus, 2018; Kallus & Zhou, 2018a; Athey & Wager, 2021; Hatt et al., 2022) and is common for decision-making in a wide range of practical applications such as, e.g., automated hiring, credit lending, and ad targeting (e.g., Smith et al., 2023; Yoganarasimhan et al., 2022; Kozodoi et al., 2022).

We make use of the Neyman-Rubin potential outcomes framework (Rubin, 1978) and denote $Y(a)$ as the potential outcome, which would have been observed if the action had been set to $A = a$. Formally, a policy is a measurable function $\pi \colon \mathcal{X} \times \mathcal{S} \to [0, 1]$, which maps an individual with covariates $(X, S)$ onto a probability of receiving an action. The policy value of $\pi$ is then defined as

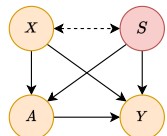

$$V(\pi) = \mathbb{E}[Y^\pi] = \mathbb{E}[\pi(X, S)\, Y(1) + (1 - \pi(X, S))\, Y(0)]. \tag{1}$$

Note that we cannot directly estimate the policy value because, for each observation, only one of the potential outcomes is observed in the observational data. This is known as the fundamental problem of causal inference (Pearl, 2009). However, we can impose the following standard assumptions in order to identify the policy value $V(\pi)$ from observational data (Rubin, 1974).

Figure 1: Causal graph. We allow for arbitrary dependence between $X$ and $S$.

**Assumption 1** (Standard causal inference assumptions). We assume: (i) *consistency:* $Y(A) = Y$; (ii) *positivity:* $0 < \mathbb{P}(A = 1 \mid X = x, S = s) < 1$ for all $x \in \mathcal{X}$; and (iii) *strong ignorability:* $Y(0), Y(1) \perp\!\!\!\perp A \mid X$.

Under Assumption 1, the policy value is identified by $V(\pi) = \mathbb{E}_W[\psi^{\mathrm{m}}(\pi, W)]$, with observational data $W = (X, S, A, Y)$ and where $\psi^{\mathrm{m}}(\pi, W)$ is one of the following three policy scores:

$$\psi^{\mathrm{DM}}(\pi, W) = \pi(X, S)\, \mu_1(X, S) + (1 - \pi(X, S))\, \mu_0(X, S), \tag{2}$$

$$\psi^{\mathrm{IPW}}(\pi, W) = \frac{A\, \pi(X, S) + (1 - A)\,(1 - \pi(X, S))}{A\, \pi_b(X, S) + (1 - A)\,(1 - \pi_b(X, S))} Y, \qquad \text{and} \tag{3}$$

$$\psi^{\mathrm{DR}}(\pi, W) = \psi^{\mathrm{DM}}(\pi, W) + \frac{A\, \pi(X, S) + (1 - A)\,(1 - \pi(X, S))}{A\, \pi_b(X, S) + (1 - A)\,(1 - \pi_b(X, S))} (Y - \mu_A(X, S)), \tag{4}$$

which refer to the direct method (DM), the inverse propensity score weighted (IPW) method, and the doubly robust (DR) method, and where $\mu_j(X, S) = \mathbb{E}[Y \mid X, S, A = j], j \in \{0, 1\}$, are the

---

[3]In the literature on causal machine learning, actions are oftentimes also called treatments (e.g., Curth & van der Schaar, 2021). Throughout our manuscript, we prefer the term "action" as it directly relates to the decision-making literature.

response surfaces and where $\pi_b(X, S) = \mathbb{P}(A = 1 \mid X, S)$ is the propensity score (i.e., behavioral policy). Both $\mu_j(X, S)$ and $\mu_j(X, S)$ are also called nuisance parameters. Both are ground-truth components for the data-generating process which can be estimated from the observational data.

**Task:** In standard off-policy learning, the objective is to find a policy from observational data that maximizes the policy value via

$$\pi^{\mathrm{uf}} \in \arg\max_{\pi \in \Pi} V(\pi), \tag{5}$$

where $\Pi$ is some predefined class of policies. For example, $\Pi$ may contain all policies parameterized by some neural network. Any policy that satisfies Eq. (5) is an *optimal unrestricted policy*, as it does not give any special considerations to the sensitive covariates $S$ when maximizing the policy value. In special cases, the optimal unrestricted policy may coincide with a policy that satisfies the desired fairness notion but, in practice, it will generally not. In many situations, the optimal unrestricted policy will lead to discrimination because of which fairness must be explicitly enforced.

## 4 Fairness notions for off-policy learning

We now introduce different fairness notions that are tailored to off-policy learning. Specifically, fairness may enter off-policy learning at two different stages, namely with respect to (1) the action and (2) the policy value. We refer to them as (1) *action fairness* and (2) *value fairness*, respectively. The former, action fairness, prohibits discrimination with respect to the selected action, while the latter, value fairness, prohibits discrimination with respect to the expected utility (i.e., the policy value).

∎**Action fairness:** The objective in action fairness is that the prediction of a policy should not depend on an individual's sensitive attributes. For example, in credit lending, credit approval should not be dependent on the gender of the applicants. We formalize this in the following definition.

**Definition 1** (Action fairness). *A policy $\pi^{\mathrm{af}} \in \Pi$ fulfills action fairness if it is not a function of $S$ and $\pi^{\mathrm{af}}(X) \perp\!\!\!\perp S$, that is, the recommended action should be independent of the sensitive attribute. A policy $\pi^{\mathrm{af}}$ that fulfills action fairness is optimal if it satisfies $\pi^{\mathrm{af}} \in \arg\max_{\pi \in \Pi_{\mathrm{af}}} V(\pi)$, where $\Pi_{\mathrm{af}} = \{\pi \in \Pi \mid \pi \text{ fulfills action fairness}\}$.*

Action fairness is the equivalent of *demographic parity* for decision-making (Hardt et al., 2016). It ensures that individuals who only differ with respect to their sensitive attributes (and covariates correlated to them) receive the same decision. As such, action fairness is relevant in many applications such as hiring or credit lending where legal frameworks mandate that decisions may not discriminate against certain sensitive attributes (Barocas & Selbst, 2016; Kleinberg et al., 2019).

∎**Value fairness:** The rationale behind value fairness is that different sub-populations defined by the sensitive attribute may benefit differently from a policy. Hence, we now express fairness with respect to the policy value and thus ensure that individuals with different sensitive attributes achieve, on average, a similar utility. To formalize value fairness, let us denote the conditional policy value $V_s(\pi) = \mathbb{E}[\psi^{\mathrm{m}}(\pi, W) \mid S = s]$, where we condition on the sensitive attribute $S = s$. In the following, we introduce two variants of value fairness with different aims: (1) envy-free fairness and (2) max-min fairness. The former, envy-free fairness, ensures that the conditional policy values $V_s(\pi), s \in \{0, 1\}$, do not differ more than some predefined level $\alpha$ between the sub-populations. The latter, max-min fairness, ensures that the worst-case conditional policy value across sub-populations is being maximized.

**Definition 2** (Envy-free fairness). *A policy $\pi \in \Pi$ fulfills envy-free fairness with level $\alpha \geq 0$ if $|V_s(\pi) - V_{s'}(\pi)| \leq \alpha$ for all $s, s' \in S$. We denote the set of envy-free policies by $\Pi(\alpha) = \{\pi \in \Pi \mid \pi \text{ is envy free with level } \alpha\}$. An envy-free policy $\pi^\alpha$ is optimal if $\pi^\alpha \in \arg\max_{\pi \in \Pi(\alpha)} V(\pi)$.*

**Definition 3** (Max-min fairness). *A policy $\pi^{\mathrm{mm}} \in \Pi$ fulfills max-min fairness if it minimizes the worst-case policy value for the sensitive attributes, that is, $\pi^{\mathrm{mm}} \in \arg\max_{\pi \in \Pi} \inf_{s \in S} V_s(\pi)$.*

The above definitions of value fairness are inspired by previous literature on resource allocation (e.g., Arnsperger, 1994; Bertsimas et al., 2011), and we here adopt them here to off-policy learning, that is, learning from observational data. Envy-free fairness allows decision-makers to control for disparities in the utility between the sensitive groups by fixing $\alpha$. Max-min fairness seeks the best possible worst-case policy value.

■ **Combining action fairness and value fairness:** Both action fairness and value fairness can be combined in off-policy learning so that the obtained policies fulfill both notions simultaneously. To this end, one simply replaces the policy class $\Pi$ with $\Pi_{\text{af}}$. This thus restricts the policy class to all policies that fulfill action fairness, and, as a result, one obtains policies that fulfill both notions.

Combining action fairness and value fairness has also theoretical implications, which we discuss in the following. In fact, it turns out that the notion of max-min fairness only yields a useful fairness notion when it is used in combination with action fairness. We show this in the following Lemma 1.

**Lemma 1.** *Let $\Pi$ the set of all measurable policies $\pi\colon \mathcal{X} \times \mathcal{S} \to [0,1]$. Then, there exists a policy that fulfills max-min fairness, i.e., $\pi^* \in \arg\max_{\pi \in \Pi} \inf_{s \in S} V_s(\pi)$ which is also an optimal unrestricted policy (i.e., a solution to Eq. (5)).*

We now turn to the relationship between envy-free fairness and max-min fairness when combined with action fairness. As it turns out, under action fairness and some further conditions, max-min fairness can be seen as a special case of envy-free fairness with $\alpha = 0$. This is stated in Lemma 2. We provide an additional discussion of the assumptions from Lemma 2 in Appendix D. Furthermore, we provide a toy example to discuss our fairness notions in Appenidx C.

**Lemma 2.** *Let $\text{ITE}(x, s^*) = \mu_1(x, s^*) - \mu_0(x, s^*)$ denote the individual treatment effect for an individual with covariates $(x, s^*)$. We further assume that $\mathcal{S} = \{0, 1\}$ is binary, and let $(\pi^{\text{mm}}, s^*)$ be a solution to $\max_{\pi \in \Pi} \inf_{s \in S} V_s(\pi)$, and let $\pi^{\text{mm}}(x)$ fulfill action fairness. Furthermore, we assume that there exists a set of covariates $V \subseteq \mathcal{X}$ with $\mathbb{P}(X \in V \mid S = s) > 0$ such that: either $\text{ITE}(x, s^*) > 0$ and $\pi^{\text{mm}}(x) < 1$; or $\text{ITE}(x, s^*) < 0$ and $\pi^{\text{mm}}(x) > 0$ for all $x \in V$, $s \in \mathcal{S}$. Then, $\pi^{\text{mm}}$ fulfills envy-free fairness with $\alpha = 0$. Further, all optimal policies that both satisfy action fairness and envy-free fairness with $\alpha = 0$ also fulfill max-min fairness.*

## 5    NEURAL FRAMEWORK FOR OFF-POLICY LEARNING

We propose our neural framework, called FairPol, which learns optimal action and/ or value fair policies in two steps (see Fig. 2). In **Step 1**, we ensure action fairness by restricting the underlying policy class $\Pi$ to a subset of policies $\Pi_{\text{af}} \subseteq \Pi$ (Sec. 5.1). In **Step 2**, we ensure value fairness by changing the underlying learning objective (Sec. 5.2). We provide theoretical results in Sec. 5.3.

### 5.1    STEP 1: FAIR REPRESENTATION LEARNING FOR ACTION FAIRNESS

To obtain $\Pi_{\text{af}}$, we build upon the idea of fair representation learning (e.g., Zemel et al., 2013; Madras et al., 2018) but adapt it to our task of fair off-policy learning. We first learn a fair representation $\Phi\colon \mathcal{X} \to \mathbb{R}^k$ of the data so that $\Phi(X) \perp\!\!\!\perp S$, but where $\Phi(X)$ is still predictive of the outcome $Y$. This ensures that any policy based on $\Phi(X)$ satisfies action fairness but is still effective in achieving a large policy value. In our implementation, we parameterize $\Phi$ by neural networks that are trained with two adversarial objectives. As a result, $\Phi$ essentially yields a policy class that is restricted to all policies with action fairness, that is, $\Pi_{\text{af}}^{\Phi} = \{\pi_\theta \circ \Phi \mid \theta \in \Theta\}$.

We use three feed-forward neural networks to learn the representation $\Phi$: (1) a base representation network $\Phi_{\theta_\Phi}$ that takes the non-sensitive attributes $X$ as input and outputs the representation; (2) an outcome prediction network $G_{\theta_Y}^Y$ that predicts the outcome $Y$ based on the representation $\Phi$; and (3) a sensitive attribute network $G_{\theta_S}^S$ that predicts the sensitive attribute $S$ based on the representation. Here $\theta_\Phi$, $\theta_Y$, and $\theta_S$ denote the neural network parameters. The base representation network $\Phi_{\theta_\Phi}$ serves as basis to construct the fair representation, while $G_{\theta_Y}^Y$ and $G_{\theta_S}^S$ allow us to ensure predictiveness of $Y$ and non-predictiveness of $S$.

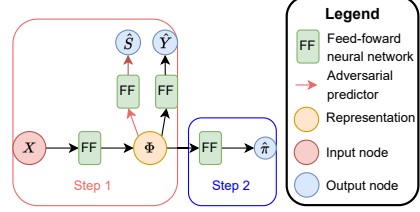

Figure 2: Overview of FairPol which provides an instantiation of our framework with neural networks.

We proceed as follows to find the optimal parameters $\hat{\theta}_\Phi$, $\hat{\theta}_Y$, and $\hat{\theta}_S$. We optimize an objective consisting of three parts: (1) The outcome loss $\mathcal{L}_Y$ ensures that to our representation $\Phi$ and the outcome prediction network are predictive of the outcome $Y$. For this, we mini-

mize $\mathcal{L}_Y(\theta_\Phi, \theta_Y) = \frac{1}{n} \sum_{i=1}^{n} \left( G_{\theta_Y}^Y \left( \Phi_{\theta_\Phi}(X_i) \right) - Y_i \right)^2$. (2) The sensitivity loss $\mathcal{L}_S$ learns the parameters of the sensitive attribute network, i.e., $G_{\theta_S}^S$, so that it is predictive of $S$. We thus minimize $\mathcal{L}_S(\theta_\Phi, \theta_S) = \frac{1}{n} \sum_{i=1}^{n} \mathrm{CE} \left( G_{\theta_S}^S \left( \Phi_{\theta_\Phi}(X_i) \right), S_i \right)$, where CE denotes the categorical cross-entropy loss. (3) The confusion loss $\mathcal{L}_{\mathrm{conf}}$, guided by the sensitive attribute network, aims to render the representation $\Phi$ non-predictive of $S$. We thus minimize $\mathcal{L}_{\mathrm{conf}}(\theta_\Phi, \theta_S) = \frac{1}{n} \sum_{i=1}^{n} \sum_{j=1}^{|\mathcal{S}|} -\frac{1}{|\mathcal{S}|} \log \left( G_{\theta_S}^S [\Phi_{\theta_\Phi}(X_i)]^j \right)$, where $[\cdot]^j$ is the $j$-th element of a vector.

Both the sensitivity loss and the confusion loss are adversarial to each other. This is crucial for the following reasons: the sensitive attribute network $G_{\theta_S}^S$ is trained to correctly classify the sensitive attribute by minimizing $\mathcal{L}_S(\theta_\Phi, \theta_S)$ with respect to $\theta_S$, while the base representation network $\Phi_{\theta_\Phi}$ tries to "confuse" the sensitive attribute network by minimizing $\mathcal{L}_{\mathrm{conf}}(\theta_\Phi, \theta_S)$ with respect to $\theta_\Phi$, i.e., forcing the sensitive attribute network to predict a uniform distribution of the sensitive attribute. This ensures that the learned representation becomes non-predictive of the sensitive attribute $S$. Taken together, the overall objective is

$$\hat{\theta}_\Phi, \hat{\theta}_Y = \arg\min_{\theta_\Phi, \theta_Y} \mathcal{L}_Y(\theta_\Phi, \theta_Y) + \gamma \mathcal{L}_{\mathrm{conf}}(\theta_\Phi, \hat{\theta}_S); \qquad \hat{\theta}_S = \arg\min_{\theta_S} \gamma \mathcal{L}_S(\hat{\theta}_\Phi, \theta_S); \quad (6)$$

where $\gamma$ is a parameter that weights the different parts in the loss function. The objective in Eq. (6) is also known as counterfactual domain confusion loss (Melnychuk et al., 2022). We later train the two adversarial objectives from Eq. (6) via an iterative gradient-based solver. For further details on our learning algorithm, we refer to Appendix E.

## 5.2 Step 2: Learning objectives for value fairness

We now address how we incorporate fairness in off-policy learning, i.e., specify the learning objectives in our framework and how these vary according to the different notions of value fairness. To do so, we first propose model-agnostic objectives and then describe how we incorporate these into Step 2 of FairPol.

**Model-agnostic objectives**: In expectation, the policy value is defined as $V(\pi) = \mathbb{E}_W[\psi^{\mathrm{m}}(\pi, W)]$, where $\mathrm{m} \in \{\mathrm{DM}, \mathrm{IPW}, \mathrm{DR}\}$. Further, the conditional policy value is defined as $V_s(\pi) = \mathbb{E}[\psi^{\mathrm{m}}(\pi, W) \mid S = s] = \mathbb{E}[\psi^{\mathrm{m}}(\pi, W) \frac{\mathbb{1}(S=s)}{\mathbb{P}(S=s)}]$, where $\mathbb{1}(\cdot)$ denotes the indicator function. Hence, we can estimate these quantities by replacing the expectations with finite sample averages. Then, the empirical policy value becomes $\hat{V}^{\mathrm{m}}(\pi) = \frac{1}{n} \sum_{i=1}^{n} \psi^{\mathrm{m}}(\pi, W_i)$, , and the empirical conditional policy value becomes $\hat{V}_s^{\mathrm{m}}(\pi) = \frac{1}{n} \sum_{i=1}^{n} \frac{\mathbb{1}(S_i=s)}{\hat{p}_n(s)} \psi^{\mathrm{m}}(\pi, W_i)$ with $\hat{p}_n(s) = \frac{\sum_{j=1}^{n} \mathbb{1}(S_j=s)}{n}$. The optimal unconstrained policy can be obtained via $\hat{\pi} \in \arg\max_{\pi \in \Pi} \hat{V}^{\mathrm{m}}(\pi)$.

We now state the learning objectives for (1) envy-free fairness and (2) max-min fairness: (1) For envy-free fairness, we reformulate the optimization problem over the class of envy-free policies into an optimization problem over an unconstrained policy class. We further replace the population quantities $V(\pi)$ and $V_s(\pi)$ with their corresponding estimates $\hat{V}^{\mathrm{m}}(\pi)$ and $\hat{V}_s^{\mathrm{m}}(\pi)$. We thus yield $\hat{\pi}^\lambda \in \arg\max_{\pi \in \Pi} \hat{V}_\lambda^{\mathrm{m}}(\pi)$ with $\hat{V}_\lambda^{\mathrm{m}}(\pi) = \hat{V}^{\mathrm{m}}(\pi) - \lambda \max_{s, s' \in \mathcal{S}} |\hat{V}_s^{\mathrm{m}}(\pi) - \hat{V}_{s'}^{\mathrm{m}}(\pi)|$, where $\lambda > 0$ is a hyperparameter controlling envy-free fairness and where larger values correspond to more fair policies. (2) For max-min fairness, we proceed analogously and obtain $\hat{\pi}^{\mathrm{mm}} \in \arg\max_{\pi \in \Pi} \min_{s \in S} \hat{V}_s^{\mathrm{m}}(\pi)$.

**Incorporating value fairness in FairPol:** The second step of FairPol is to optimize the empirical policy value. Here, we optimize against the previously introduced learning objectives. Depending on whether action fairness is enforced, we optimize the learning objectives over all policies in $\Pi$ or the subset $\Pi_{\mathrm{af}}^\Phi$ of policies with action fairness. Hence, once the representation $\hat{\Phi} = \Phi_{\hat{\theta}_\Phi}$ is trained, we optimize our objectives for value fairness over the policy class $\Pi_{\mathrm{af}}^{\hat{\Phi}} = \{\pi_\theta \circ \hat{\Phi} \mid \theta \in \Theta\}$. Here, we parametrize $\pi_\theta$ by a neural network with parameters $\theta \in \Theta$ that takes the representation $\hat{\Phi}(X)$ as input and outputs a policy recommendation $\pi_\theta(\hat{\Phi}(X)) \in [0, 1]$. Formally, we thus optimize the policy via $\max_{\theta \in \Theta} \hat{V}^{\mathrm{m}}(\pi_\theta)$, $\max_{\theta \in \Theta} \hat{V}_\lambda^{\mathrm{m}}(\pi_\theta)$, or $\max_{\theta \in \Theta} \min_{s \in S} \hat{V}_s^{\mathrm{m}}(\pi_\theta)$, depending on whether there is no value fairness, envy-free fairness, or max-min fairness, respectively.[4]

---

[4]Note that the policies that fulfill no value fairness are either the optimal unrestricted policies or the policies that fulfill action fairness.

**Implementation details:** In our FairPol implementation, we use feed-forward neural networks with dropout and exponential linear unit activation functions for the base representation network, the outcome prediction network, and the sensitive attribute network. We use Adam (Kingma & Ba, 2015) for the optimization in both Steps 1 and 2. We further follow best practices for hyperparameter tuning. We first split the data into a training and validation set, and we then perform hyperparameter tuning using a grid search. All evaluations are based on the test set so that we capture the out-of-sample performance on unseen data. Additional details for our framework are in Appendix F.

### 5.3 GENERALIZATION BOUNDS

We derive generalization bounds for the finite-sample version of our framework under the following standard boundedness assumption.

**Assumption 2** (Boundedness). We assume there exist constants $C, \eta, \nu > 0$ such that (i) the outcomes are bounded with $|Y| \leq C$ almost surely, (ii) the propensity score is bounded away from 0 and 1, i.e., $\mathbb{P}(\eta \leq \pi_b(X, S) \leq 1 - \eta) = 1$, and (iii) $S$ has full support on $\mathcal{S}$, i.e., $p(s) = \mathbb{P}(S = s) \geq \nu$ for all $s \in \mathcal{S}$ and some $\nu > 0$.

The following result quantifies the deviation of the proposed finite-sample policy estimators from their respective population quantities. Note that the derivations also hold for action fairness where one would simply need to replace $\Pi$ by $\Pi_{\mathrm{af}}$.

**Theorem 1** (Generalization bounds). *Let $p(s) = \mathbb{P}(S = s) \geq \nu$ for all $s \in \mathcal{S}$ and some $\nu > 0$. Let $p, p_1, p_2 > 0$ and let $K_{\mathrm{m}}$ denote a constant that depends on the estimation method $\mathrm{m} \in \{\mathrm{DM}, \mathrm{IPW}, \mathrm{DR}\}$ as follows: $K_{\mathrm{DM}} = 1$, $K_{\mathrm{IPW}} = \frac{1}{2\eta}$, and $K_{\mathrm{DR}} = \frac{\eta+1}{\eta}$. Assume that, for $\ell(n, p_2) = 1 - \nu + \sqrt{\log(|\mathcal{S}|/p_2)/2}$, it holds that $\frac{1}{\sqrt{n}}\ell(n, p_2) < \nu$. Then, the following three statements hold: (i) With probability at least $1 - p$ it holds that*

$$V(\pi) \geq \hat{V}^{\mathrm{m}}(\pi) - 2CK_{\mathrm{m}}\left(R_n(\Pi) + \sqrt{\frac{8\log\left(\frac{2}{p}\right)}{n}}\right) \tag{7}$$

*for all $\pi \in \Pi$. (ii) With probability at least $1 - p_1 - p_2$, we have*

$$V_\lambda(\pi) \geq \hat{V}_\lambda^{\mathrm{m}}(\pi) - 2CK_{\mathrm{m}}\frac{2+\nu}{\nu}\left(R_n(\Pi) + \sqrt{\frac{8\log\left(\frac{4|\mathcal{S}|}{p_1}\right)}{n}} + \frac{2}{(2+\nu)\sqrt{n}}\left(\frac{\ell(n,p_2)}{\nu - \frac{1}{\sqrt{n}}\ell(n,p_2)}\right)\right) \tag{8}$$

*for all $\pi \in \Pi$. (iii) With probability at least $1 - p_1 - p_2$ it holds that*

$$\min_{s \in \mathcal{S}} V_s(\pi) \geq \min_{s \in \mathcal{S}} \hat{V}_s^{\mathrm{m}}(\pi) - \frac{2CK_{\mathrm{m}}}{\nu}\left(R_n(\Pi) + \sqrt{\frac{8\log\left(\frac{2|\mathcal{S}|}{p_1}\right)}{n}} + \frac{1}{\sqrt{n}}\left(\frac{\ell(n,p_2)}{\nu - \frac{1}{\sqrt{n}}\ell(n,p_2)}\right)\right) \tag{9}$$

*for all $\pi \in \Pi$.*

Theorem 1 shows that, with sufficient sample size, the oracle policy objectives $\hat{V}^{\mathrm{m}}(\pi)$, $\hat{V}_\lambda^{\mathrm{m}}(\pi)$, and $\min_{s \in \mathcal{S}} V_s(\pi)$ are with high probability lower bounded than their empirical counterpart if the policy class $\Pi$ has as vanishing Rademacher complexity $R_n(\Pi)$. Theorem 1 has two main qualitative implications: (i) We can achieve a $1/\sqrt{n}$-convergence rate for all fairness objectives whenever we optimize over a model class $\Pi$ with $\sqrt{n}$-vanishing Rademacher complexity $R_n(\Pi) = \mathcal{O}\left(n^{-1/2}\right)$, such as neural networks. (ii) Compared to the bound for the unrestricted policy (Eq. (7)), the bounds corresponding to our fairness objectives depend on $\nu$ and hence on the population balance within the marginal distribution of the sensitive attribute $S$. This implies that the bounds become loose whenever a sensitive group is underrepresented in the data.

## 6 EXPERIMENTS

**Experimental setup using simulated data:** We generate a simulated dataset with $n = 3000$ observations inspired by a credit-lending problem (see Appendix G for details). Throughout our experiments, we estimate the policies using the data from a training set (80%) and evaluate the policies using the data from a test set (20%) to compare the out-of-sample performance. We perform all evaluations using the following performance metrics: (1) We report the policy value $\hat{V}^{\mathrm{m}}(\pi)$. This thus corresponds to the objective function in off-policy learning that is maximized under the fairness constraints (thus: larger values are better). (2) We additionally report the policy value by different

sub-populations given by the conditional policy value $\hat{V}_s^{\mathrm{m}}(\pi)$ for $s = 0$ and $s = 1$. We provide the results for our framework across all three different policy scores, namely $\mathrm{m} \in \{\mathrm{DM}, \mathrm{IPW}, \mathrm{DR}\}$ from Eq. (2), Eq. (3), and Eq. (4), respectively. Of note, we cannot compare our framework against other methods since suitable baselines that can deal with fair off-policy learning over general policies are missing.

Table 1: Results for simulated data.

**Results for action fairness:** We now examine whether our framework is effective in learning policies that fulfill action fairness. (1) We first report an optimal unrestricted policy that has access to the ground-truth outcome functions from the data-generating process and acts as the maximum achievable policy value for comparison.

| Approach | Policy value | | | Action fairness |
|---|---|---|---|---|
| | Overall | $S = 0$ | $S = 1$ | |
| BASELINES | | | | |
| Optimal unrestricted policy | $1.24 \pm 0.03$ | $0.74 \pm 0.03$ | $1.46 \pm 0.06$ | $2.42 \pm 0.20$ |
| Oracle action fairness | $1.03 \pm 0.02$ | $0.01 \pm 0.07$ | $1.46 \pm 0.06$ | $0.00 \pm 0.00$ |
| OUR FAIRPOL (ONLY ACTION FAIRNESS) | | | | |
| FairPol with m = DM | $1.02 \pm 0.02$ | $0.01 \pm 0.06$ | $1.45 \pm 0.06$ | $0.21 \pm 0.05$ |
| FairPol with m = IPW | $1.01 \pm 0.03$ | $0.02 \pm 0.05$ | $1.43 \pm 0.07$ | $0.24 \pm 0.06$ |
| FairPol with m = DR | $1.01 \pm 0.03$ | $0.02 \pm 0.05$ | $1.43 \pm 0.07$ | $0.23 \pm 0.05$ |
| OUR FAIRPOL WITH ENVY-FREE FAIRNESS | | | | |
| FairPol with m = DM | $0.87 \pm 0.17$ | $0.61 \pm 0.07$ | $0.99 \pm 0.24$ | $0.38 \pm 0.22$ |
| FairPol with m = IPW | $0.87 \pm 0.05$ | $0.32 \pm 0.17$ | $1.11 \pm 0.14$ | $0.79 \pm 0.30$ |
| FairPol with m = DR | $0.86 \pm 0.06$ | $0.34 \pm 0.17$ | $1.09 \pm 0.15$ | $0.75 \pm 0.32$ |
| OUR FAIRPOL WITH MAX-MIN FAIRNESS | | | | |
| FairPol with m = DM | $0.73 \pm 0.03$ | $0.73 \pm 0.03$ | $0.73 \pm 0.03$ | $0.00 \pm 0.00$ |
| FairPol with m = IPW | $0.73 \pm 0.03$ | $0.73 \pm 0.03$ | $0.73 \pm 0.03$ | $0.00 \pm 0.01$ |
| FairPol with m = DR | $0.73 \pm 0.03$ | $0.73 \pm 0.03$ | $0.73 \pm 0.03$ | $0.00 \pm 0.01$ |

Reported: mean $\pm$ standard deviation ($\times 10$) on test set over 5 runs.

(2) We further estimate an oracle policy that fulfills action fairness with access to the ground-truth outcome functions. It should be regarded as an upper bound for the policy value that can be achieved under action fairness. (3) We compare our FairPol for action fairness, setting $\gamma = 0.5$. We report three different variants of our FairPol by varying the underlying policies scores m, namely DM, IPW, and DR.

The results are in Table 1. Besides policy values, we also report the performance in terms of action fairness, which we calculate via $\mathbb{E}[\pi(X_u, X_{s=1}, S = 1) - \pi(X_u, X_{s=0}, S = 0)]$. We make the following observations. First, the optimal unrestricted policy has the largest policy value but fails to achieve action fairness, as expected. Second, the policy value for the oracle policy with action fairness is lower, and, by definition, the action fairness achieves a score of zero. Third, we find that our FairPol is effective in achieving action fairness. Fourth, we find that our FairPol attains a policy value that is close to the upper bound given by the oracle policy with action fairness, which corroborates the effectiveness of our framework. Finally, we find that our FairPol achieves a similar performance regardless of the underlying policy score (i.e., DM, IPW, and DR) and thus appears robust with respect to the choice of policy score.

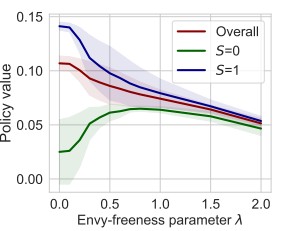 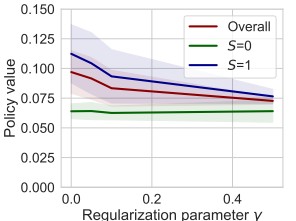

**Results for value fairness:** We further assess whether our framework is effective in learning policies that fulfill value fairness. Here, we report results from our framework with action fairness for three different fairness notions: (1) no value fairness, (2) envy-free fairness ($\lambda = 0.5$), and (3) max-min fairness. We set $\gamma = 0.5$ and provide a sensitivity analysis for the parameter later. The results are in

Figure 3: Sensitivity analysis for the envy-free parameter $\lambda$ (left) and the regularization parameter $\gamma$ (right).

Table 1. We arrive at the following conclusions. First, the optimal unrestricted policy has the largest overall policy value, as expected. Second, our FairPol for envy-free fairness achieves a smaller overall policy due to the fairness constraints. Third, our FairPol for max-min fairness is effective in achieving the desired fairness notion. It achieves a larger worst-case policy value compared to the optimal unrestricted policy and a lower difference between groups. In summary, the experimental results confirm the effectiveness of our empirical framework in enforcing the proposed fairness notions.

We also examine the sensitivity to the envy-freeness parameter $\lambda$. We compare the policy value from our FairPol for different values of $\lambda \in [0, 2]$ and choose $m = \mathrm{DR}$. The results are shown in Fig. 3 (left). As expected, the policy value decreases and the difference between the policy values for the two sub-groups $S \in \{0, 1\}$ becomes smaller for larger $\lambda$. Furthermore, the results remain robust for different choices of $\gamma$ (Fig. 3, right).

**Experimental setup using real-world data:** We now demonstrate the applicability of our framework to real-world, medical data. We use medical data from the Oregon health insur-

Table 2: Results for real-world data.

| Approach | Policy value | | | Action fairness |
|---|---|---|---|---|
| | Overall | $S$ = male | $S$ = female | |
| Optimal unrestricted policy | $0.137 \pm 0.005$ | $0.101 \pm 0.008$ | $0.165 \pm 0.003$ | $0.129 \pm 0.007$ |
| FairPol (only action fairness) | $0.130 \pm 0.004$ | $0.093 \pm 0.006$ | $0.160 \pm 0.003$ | $0.015 \pm 0.001$ |
| FairPol with envy-free fairness | $0.130 \pm 0.004$ | $0.093 \pm 0.006$ | $0.160 \pm 0.003$ | $0.067 \pm 0.007$ |
| FairPol with max-min fairness | $0.131 \pm 0.008$ | $0.100 \pm 0.011$ | $0.157 \pm 0.008$ | $0.057 \pm 0.010$ |

Reported: mean $\pm$ standard deviation on test set over 5 random runs

ance experiment (Finkelstein et al., 2012). The Oregon health insurance experiment took place in 2008. As part of it, around 30,000 low-income, uninsured adults in Oregon were offered free health insurance through Medicaid. We use our framework to learn fair policies that assign Medicaid to minimize the total costs for medical care of an individual, while avoiding discrimination with respect to gender. Besides gender, we include five additional variables as possible confounders. Details are Appendix H. FairPol is based on $\gamma = 0.5$ (for action fairness) and the double robust method m = DR. For envy-free fairness, we set $\lambda = 0.3$. Here, we do not know the ground-truth outcomes for real-world data, and, hence, we estimate the nuisance parameters using a TARNet (Shalit et al., 2017). We then estimate the (conditional) policy values on the test data using the estimators from Sec. 5.2. To quantify action fairness, we report Spearman's rank correlation coefficient between the sensitive attribute (gender) and the policy predictions on the test data. For details, we refer to Appendix H.

**Results for action and value fairness:** The results are shown in Table 2. Again, the optimal unrestricted policy has the largest empirical policy value but does not satisfy action fairness. FairPol with only action fairness is effective at enforcing the desired fairness notion, but this comes at the cost of a slightly worse policy value. However, this is to be expected as enforcing action fairness can worsen value fairness (we refer to our toy example in Appendix C for a detailed discussion on the tradeoff). Furthermore, FairPol with max-min fairness is effective in improving value fairness compared to FairPol with only action fairness. In summary, the results on real-world data confirm the effectiveness of our framework.

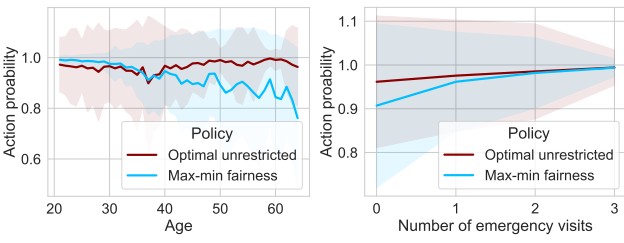

Figure 4: Comparison of the estimated policies averaged over 20 random runs. Visualized are the policy predictions (i.e., the outputs of the respective policies).

**Insights:** We now examine the outputs of the respective policies. We the averaged outputs over (i) age and (ii) the number of previous emergency department visits of an individual (Fig. 4). Both policies tend to recommend Medicaid for the majority of individuals. Furthermore, both policy outputs are lower for individuals with a smaller number of emergency department visits. This is reasonable as such individuals may be less at risk of accumulating medical debt compared to individuals with an extensive medical history. FairPol with max-min fairness outputs slightly lower predictions for older individuals and for individuals with no or few emergency visits. Hence, there seem to exist some male individuals with few emergency visits or higher age for which free health insurance has only little positive effect.

**Discussion:** In this paper, we proposed a novel framework for fair off-policy learning from observational data. For this, we introduced fairness notions tailored to off-policy learning, then developed a flexible instantiation of our framework using machine learning (FairPol), and finally provided theoretical guarantees in the form of generalization bounds. Our framework is widely applicable to algorithmic decision-making in practice where fairness must be ensured. In Appendix J, we discuss the pros/cons of the different fairness criteria, discuss suitable use cases, and provide recommendations for practice.

Our work contributes to the literature in the following ways: (i) We integrate fairness notions such as envy-free fairness or max-min fairness (which have been used in traditional, utility-based decision models) into off-policy learning. (ii) We provide a theoretical understanding of how these fairness notions interact in the context of off-policy learning. (iii) We propose a practical framework to learn fair optimal policies and provide theoretical guarantees in the form of generalization bounds. For future work, it may be interesting to consider different fairness notions and/or off-policy learning settings, e.g., when treatments are assigned over time.

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

# A EXTENDED RELATED WORK

## A.1 ALGORITHMIC FAIRNESS FOR MACHINE LEARNING PREDICTIONS

Extensive work has developed algorithmic fairness for machine learning predictions, which refers to computational approaches that enforce certain constraints on predictions so that similarly situated individuals also receive similar predictions. In the following, we provide a brief overview of the different concepts and fairness notions. We refer to Chouldechova & Roth (2020) and Mitchell et al. (2021) for a detailed overview. We emphasize that the following fairness notions are developed for predictions and *not* for off-policy learning.

A major literature branch deals with fairness notions that prevent systematic differences in predictions across different groups that are defined by some sensitive attributes (e.g., gender or race) (e.g., Dwork et al., 2012; Hardt et al., 2016; Madras et al., 2018; Corbett-Davies et al., 2023).This can be achieved, for example, by enforcing independence between the sensitive attribute and the predictions (i.e., statistical parity) or ensuring a similar classification performance for the different sensitive groups (e.g., similar type-I/II error rates). Approaches for group-level fairness have been extended to specific settings, such as for data with unobserved sensitive attributes (Kallus et al., 2021) and for censored training data (Kallus & Zhou, 2018b). Beyond group-level fairness, there are also notions at the individual level as well as notions that are based on a causal lens (called causal fairness); see Chouldechova & Roth (2020). Note that, even though off-policy learning itself is a causal problem, our setting later is different from the literature on causal fairness: the standard literature on causal fairness uses causal theory (e.g., structural causal models) to define fairness notions (e.g., Kilbertus et al., 2017; Kusner et al., 2017; Nabi & Shpitser, 2018), while we introduce fairness to a specific causal decision problem (off-policy learning).

## A.2 ALGORITHMIC FAIRNESS FOR UTILITY-BASED DECISION MODELS

A different literature stream has developed fairness notions that account for the utility of individuals who are subject to decisions. Such fairness notions have been integrated into traditional decision problems and thus outside of machine learning. Examples are, for instance, resource allocation (Bertsimas et al., 2011; 2013; Rea et al., 2021) and pricing (Kallus & Zhou, 2021; Cohen et al., 2022). Here, a common fairness notion is envy-free fairness, which is fulfilled if an individual receives an allocation that has the same (or a higher) utility as the allocation of any other individual. Hence, decisions are envy-free if all players receive a share of resources that is equally good from their perspective (Arnsperger, 1994). Another fairness notion is max-min fairness, which is grounded in Rawlsian justice and which seeks to maximize the minimum utility that a player can obtain (Bertsimas et al., 2011). However, to the best of our knowledge, the aforementioned notions – envy-free fairness and max-min fairness – have only been used for traditional resource allocations and have not yet been adapted to off-policy learning from observational data, which is one of our contributions later.

Prior literature also considered algorithmic fairness in specialized settings. Examples are ranking tasks such as from recommender systems (e.g., Singh & Joachims, 2019) or risk-averse approaches to bound worst-case outcomes (e.g., Fang et al., 2022). Even others consider algorithmic fairness in reinforcement learning. Here, fair policies can be obtained by customizing the reward function (Jabbari et al., 2017; Jiang & Lu, 2019; Yu et al., 2022) or by optimizing social welfare functions (Siddique et al., 2020; Zimmer et al., 2021). However, these works focus on Markov decision processes (MDPs), whereas we focus on learning policies in non-sequential settings that are not restricted to MDPs.

## A.3 CAUSAL SCM-BASED FAIRNESS FOR OFF-POLICY LEARNING

This literature stream rests on the assumption that the structural causal graph of the decision problem is known and then seeks to block specific causal pathways that are deemed unfair (Nabi et al., 2019; Nilforoshan et al., 2022). However, approaches from this literature stream require *exact knowledge* of the causal structure of the decision problem. That is, the underlying structural causal model of the data-generating process must be a prior known. In contrast to that, we do *not* make such strong

assumption (i.e., exact knowledge of the underlying causal structure) as this is rarely the case in practice.

# B PROOFS

## B.1 PROOF OF LEMMA 1

*Proof.* For each sensitive attribute $s \in \mathcal{S}$, we construct $\pi^*(\cdot, s) \in \arg\max_{\pi \in \Pi_{\mathcal{X}}} V_s(\pi)$, where $\Pi_{\mathcal{X}} = \{\pi\colon \mathcal{X} \to [0,1] \mid \pi \text{ measurable}\}$. By definition, it holds that $V_s(\pi) \leq V_s(\pi^*)$ for any policy $\pi \in \Pi$ and, hence, $\inf_{s \in S} V_s(\pi) \leq \inf_{s \in S} V_s(\pi^*)$, which means that $\pi^*$ is a policy fulfilling max-min fairness. At the same time, due to $V_s(\pi) \leq V_s(\pi^*)$, it holds that

$$V(\pi) = \int_{\mathcal{S}} V_s(\pi)\, \mathbb{P}(S = s)\, \mathrm{d}s \leq \int_{\mathcal{S}} V_s(\pi^*)\, \mathbb{P}(S = s)\, \mathrm{d}s = V(\pi^*). \tag{10}$$

Thus, the policy $\pi^*$ is an optimal unrestricted policy. □

## B.2 PROOF OF LEMMA 2

*Proof.* We first show that $V_0(\pi^{\mathrm{mm}}) = V_1(\pi^{\mathrm{mm}})$, i.e., $\pi^{\mathrm{mm}}$ is envy-free with $\alpha = 0$. Let us assume w.l.o.g. that $V_0(\pi^{\mathrm{mm}}) < V_1(\pi^{\mathrm{mm}})$. By our assumption, we can find an $\epsilon > 0$ such the policy $\pi'$ defined by

$$\pi'(x) = \begin{cases} \pi^{\mathrm{mm}}(x) + \epsilon\, \mathrm{sign}\{\mathrm{ITE}(x,0)\}, & \text{if } x \in V, \\ \pi^{\mathrm{mm}}(x), & \text{if } x \in \mathcal{X} \setminus V, \end{cases} \tag{11}$$

satisfies $V_1(\pi') > V_0(\pi^{\mathrm{mm}})$. By construction of $\pi'$ and our assumption, we yield

$$V_0(\pi') = \int_{\mathcal{X}} \pi'(x)\, \mathrm{ITE}(x,0)\, \mathbb{P}(x \mid S = 0) + \mu_0(x,0)\, \mathbb{P}(x \mid S = 0)\, \mathrm{d}x \tag{12}$$

$$> \int_{\mathcal{X}} \pi^{\mathrm{mm}}(x)\, \mathrm{ITE}(x,0)\, \mathbb{P}(x \mid S = 0) + \mu_0(x,0)\, \mathbb{P}(x \mid S = 0)\, \mathrm{d}x = V_0(\pi^{\mathrm{mm}}). \tag{13}$$

This implies

$$\min\{V_0(\pi'), V_1(\pi')\} > \min\{V_0(\pi^{\mathrm{mm}}), V_1(\pi^{\mathrm{mm}})\}, \tag{14}$$

which is a contradiction to the assumption that $\pi^{\mathrm{mm}}$ fulfills max-min fairness. Hence, $\pi^{\mathrm{mm}}$ fulfills envy-free fairness.

Now, let $\pi^0$ be an optimal policy that satisfies both action fairness and envy-free fairness. Let us further assume that $\pi^0$ does not fulfill max-min fairness. We then yield

$$V(\pi^0) = \mathbb{P}(S = 0)\, V_0(\pi^0) + \mathbb{P}(S = 1)\, V_1(\pi^0) < \mathbb{P}(S = 0)\, V_0(\pi^{\mathrm{mm}}) + \mathbb{P}(S = 1)\, V_1(\pi^{\mathrm{mm}}) = V(\pi^{\mathrm{mm}}), \tag{15}$$

which is a contradiction because $\pi^{\mathrm{mm}}$ fulfills envy-free fairness and $\pi^0$ is optimal. □

## B.3 PROOF OF GENERALIZATION BOUNDS

In this section, we provide proof of our generalization bounds, namely Theorem 1. In our proof, we later leverage ideas from Theorem 1 in Kallus (2018); however, adapting these to our setting is not straightforward, and several additional arguments must be made. To this end, we begin with three auxiliary lemmas.

**Lemma 3.** *Let* $T^{\mathrm{m}}(s, W) = \sup_{\pi \in \Pi} |\frac{1}{n} \sum_{i=1}^{n} \frac{\mathbb{1}(S_i = s)}{p(s)} \psi^{\mathrm{m}}(\pi, W_i) - V_s(\pi)|$. *Then,* $T^{\mathrm{m}}(s, \cdot)$ *satisfies the bounded difference inequality with* $\frac{4C}{np(s)} K_{\mathrm{m}}$, *where* $K_{\mathrm{m}}$ *is a constant depending on* $\mathrm{m} \in \{\mathrm{DM}, \mathrm{IPW}, \mathrm{DR}\}$.

*Proof.* It holds that

$$|T^{\mathrm{m}}(s, W) - T^{\mathrm{m}}(s, W')| \tag{16}$$

$$\overset{(1)}{\leq} \sup_{\pi \in \Pi} \left| \left| \frac{1}{n} \sum_{i=1}^n \frac{\mathbb{1}(S_i = s)}{p(s)} \psi^{\mathrm{m}}(\pi, W_i) - V_s(\pi) \right| - \left| \frac{1}{n} \sum_{i=1}^n \frac{\mathbb{1}(S'_i = s)}{p(s)} \psi^{\mathrm{m}}(\pi, W'_i) - V_s(\pi) \right| \right| \tag{17}$$

$$\overset{(2)}{\leq} \frac{1}{np(s)} \sup_{\pi \in \Pi} \left| \sum_{i=1}^n \mathbb{1}(S_i = s)\psi^{\mathrm{m}}(\pi, W_i) - \mathbb{1}(S'_i = s)\psi^{\mathrm{m}}(\pi, W'_i) \right| \tag{18}$$

$$= \frac{1}{np(s)} \sup_{\pi \in \Pi} \left( |\psi^{\mathrm{m}}(\pi, W_j)| + |\psi^{\mathrm{m}}(\pi, W'_j)| \right), \tag{19}$$

where (1) follows from a property of the supremum and (2) follows from the reverse triangle inequality. It remains to bound $|\psi^{\mathrm{m}}(\pi, \cdot)|$ for $\mathrm{m} \in \{\mathrm{DM}, \mathrm{IPW}, \mathrm{DR}\}$. For $\mathrm{m} = \mathrm{DM}$, we get

$$\left| \psi^{\mathrm{DM}}(\pi, W_j) \right| \leq |\mu_1(X, S)| + |\mu_0(X, S)| \leq 2C. \tag{20}$$

For $\mathrm{m} = \mathrm{IPW}$, we get

$$\left| \psi^{\mathrm{IPW}}(\pi, W_j) \right| \leq \frac{|Y|}{|A\pi_b(X, S) + (1 - A)(1 - \pi_b(X, S))|} \leq \frac{C}{\eta}. \tag{21}$$

Finally, for $\mathrm{m} = \mathrm{DR}$, we get

$$\left| \psi^{\mathrm{DR}}(\pi, W_j) \right| \leq \left| \psi^{\mathrm{DM}}(\pi, W_j) \right| + \frac{|Y - \mu_A(X, S)|}{|A\pi_b(X, S) + (1 - A)(1 - \pi_b(X, S))|} \leq 2C \left( \frac{\eta + 1}{\eta} \right). \tag{22}$$

Therefore, we arrive at

$$|T^{\mathrm{m}}(s, W) - T^{\mathrm{m}}(s, W')| \leq \frac{4C}{np(s)} K_{\mathrm{m}}, \tag{23}$$

with $K_{\mathrm{DM}} = 1$, $K_{\mathrm{IPW}} = \frac{1}{2\eta}$, and $K_{\mathrm{DR}} = \frac{\eta + 1}{\eta}$. $\square$

**Lemma 4.** *With probability of at least $1 - p$, it holds that*

$$T^{\mathrm{m}}(s, W) \leq \frac{2CK_{\mathrm{m}}}{p(s)} \left( R_n(\Pi) + \sqrt{\frac{8 \log \frac{2}{p}}{n}} \right). \tag{24}$$

*Proof.* Lemma 3 allows us to apply McDiarmid's inequality, resulting in

$$\mathbb{P}\left( T^{\mathrm{m}}(s, W) - \mathbb{E}\left[ T^{\mathrm{m}}(s, W) \right] \geq \epsilon \right) \leq \exp\left( -\frac{np(s)^2\epsilon^2}{8C^2 K_{\mathrm{m}}^2} \right). \tag{25}$$

Equivalently, with probability of at least $1 - p_1$, it holds that

$$T^{\mathrm{m}}(s, W) \leq \mathbb{E}\left[ T^{\mathrm{m}}(s, W) \right] + \frac{2CK_{\mathrm{m}}}{p(s)} \sqrt{\frac{2 \log \frac{1}{p_1}}{n}}. \tag{26}$$

By a standard symmetrization argument, we yield

$$\mathbb{E}\left[ T^{\mathrm{m}}(s, W) \right] \leq \mathbb{E}\left[ \frac{1}{2^n} \sum_{\epsilon \in \{-1,1\}^n} \sup_{\pi \in \Pi} \left| \frac{1}{n} \sum_{i=1}^n \epsilon_i \frac{\mathbb{1}(S_i = s)}{p(s)} \psi^{\mathrm{m}}((\pi, W_i)) \right| \right] \leq \frac{2CK_{\mathrm{m}}}{p(s)} \mathbb{E}\left[ R_n(\Pi) \right]. \tag{27}$$

Here, the Rademacher complexity $R_n(\Pi)$ satisfies the bounded differences with $\frac{2}{n}$, and we thus obtain

$$\mathbb{P}\left( R_n(\Pi) - \mathbb{E}\left[ R_n(\Pi) \right] \leq -\epsilon \right) \leq \exp\left( -\frac{n\epsilon^2}{2} \right). \tag{28}$$

This implies that, with probability of at least $1 - p_2$, it holds that

$$\mathbb{E}\left[R_n(\Pi)\right] \leq R_n(\Pi) + \sqrt{\frac{2\log\frac{1}{p_2}}{n}}. \tag{29}$$

By setting $p_1 = p_2 = \frac{p}{2}$ and applying the union bound, we yield

$$T^{\mathrm{m}}(s, W) \leq \frac{2CK_{\mathrm{m}}}{p(s)}\left(R_n(\Pi) + \sqrt{\frac{8\log\frac{2}{p}}{n}}\right) \tag{30}$$

with probability of at least $1 - p$. $\qquad\square$

In the next step, we use Lemma 4 to derive a bound on the absolute estimation error $\left|\hat{V}_s^{\mathrm{m}}(\pi) - V_s(\pi)\right|$ that holds uniformly over all policies and sensitive attributes. This is stated in Lemma 5.

**Lemma 5.** *Let* $\ell(n, p_2) = 1 - \nu + \sqrt{\frac{\log\frac{|\mathcal{S}|}{p_2}}{2}}$. *Let us further assume that* $\frac{\ell(n,p_2)}{\sqrt{n}} < \nu$. *Then, with probability of at least* $1 - p_1 - p_2$, *it holds that*

$$\sup_{\pi\in\Pi}\sup_{s\in\mathcal{S}}\left|\hat{V}_s^{\mathrm{m}}(\pi) - V_s(\pi)\right| \leq \frac{2CK_{\mathrm{m}}}{\nu}\left(R_n(\Pi) + \sqrt{\frac{8\log\frac{2|\mathcal{S}|}{p_1}}{n}} + \frac{1}{\sqrt{n}}\left(\frac{\ell(n, p_2)}{\nu - \frac{1}{\sqrt{n}}\ell(n, p_2)}\right)\right). \tag{31}$$

*Proof.* We yield

$$\sup_{\pi\in\Pi}\left|\hat{V}_s^{\mathrm{m}}(\pi) - V_s(\pi)\right| \tag{32}$$

$$= \sup_{\pi\in\Pi}\left|\frac{1}{n}\sum_{i=1}^n\frac{\mathbb{1}(S_i = s)}{\hat{p}_n(s)}\psi^{\mathrm{m}}((\pi, W_i) - V_s(\pi)\right| \tag{33}$$

$$\leq \left|\frac{1}{\hat{p}_n(s)} - \frac{1}{p(s)}\right|\sup_{\pi\in\Pi}\left|\sum_{i=1}^n\mathbb{1}(S_i = s)\psi^{\mathrm{m}}((\pi, W_i)\right| + \sup_{\pi\in\Pi}\left|\frac{1}{n}\sum_{i=1}^n\frac{\mathbb{1}(S_i = s)}{p(s)}\psi^{\mathrm{m}}((\pi, W_i) - V_s(\pi)\right| \tag{34}$$

$$\leq \frac{2CK_{\mathrm{m}}}{p(s)}\frac{|\hat{p}_n(s) - p(s)|}{\hat{p}_n(s)} + T^{\mathrm{m}}(s, W). \tag{35}$$

The absolute difference $|\hat{p}_n(s) - p(s)|$ satisfies bounded differences with constant $\frac{1}{n}$ because

$$||\hat{p}_n(s) - p(s)| - |\hat{p}'_n(s) - p(s)|| \leq |\hat{p}_n(s) - \hat{p}'_n(s)| \leq \frac{|\mathbb{1}(S_j = s) - \mathbb{1}(S'_i = s)|}{n} \leq \frac{1}{n}. \tag{36}$$

Hence, McDiamid's inequality implies

$$\mathbb{P}\left(|\hat{p}_n(s) - p(s)| - \mathbb{E}\left[|\hat{p}_n(s) - p(s)|\right] \geq \epsilon\right) \leq \exp\left(-2n\epsilon^2\right). \tag{37}$$

Thus, with probability at least $1 - p_2$ it holds that

$$|\hat{p}_n(s) - p(s)| \leq \mathbb{E}\left[|\hat{p}_n(s) - p(s)|\right] + \sqrt{\frac{\log\frac{1}{p_2}}{2n}} \tag{38}$$

$$\overset{(1)}{\leq} \frac{1}{n}\sqrt{\mathbb{E}\left[(n\hat{p}_n(s) - np(s))^2\right]} + \sqrt{\frac{\log\frac{1}{p_2}}{2n}} \tag{39}$$

$$\overset{(2)}{=} \sqrt{\frac{p(s)\left(1 - p(s)\right)}{n}} + \sqrt{\frac{\log\frac{1}{p_2}}{2n}} = \ell(n, p_2, s) \tag{40}$$

$$\leq \frac{1}{\sqrt{n}}\left(1 - \nu + \sqrt{\frac{\log\frac{1}{p_2}}{2}}\right), \tag{41}$$

where (1) follows by applying Jensen's inequality and (2) by noting that $n\hat{p}_n(s) \sim$ Binomial$(n, p(s))$ with expectation $np(s)$ and standard deviation $\sqrt{np(s)(1-p(s))}$.

The above also implies that, with probability of at least $1 - p_2$, we obtain

$$\hat{p}_n(s) \geq p(s) - \frac{1}{\sqrt{n}}\left(1 - \nu + \sqrt{\frac{\log\frac{1}{p_2}}{2}}\right) \geq \nu - \frac{1}{\sqrt{n}}\left(1 - \nu + \sqrt{\frac{\log\frac{1}{p_2}}{2}}\right) > 0, \qquad (42)$$

whenever $\frac{1}{\sqrt{n}}\left(1 - \nu + \sqrt{\frac{\log\frac{1}{p_2}}{2}}\right) < \nu$. Putting everything together via the union bound, we obtain with probability of at least $1 - p_1 - p_2$ that

$$\sup_{\pi \in \Pi}\left|\hat{V}_s^{\mathrm{m}}(\pi) - V_s(\pi)\right| \leq \frac{2CK_{\mathrm{m}}}{\nu}\left(R_n(\Pi) + \sqrt{\frac{8\log\frac{2}{p_1}}{n}} + \frac{1}{\sqrt{n}}\left(\frac{1 - \nu + \sqrt{\frac{\log\frac{1}{p_2}}{2}}}{\nu - \frac{1}{\sqrt{n}}\left(1 - \nu + \sqrt{\frac{\log\frac{1}{p_2}}{2}}\right)}\right)\right).$$

(43)

The result follows by applying the union bound over all $s \in \mathcal{S}$. $\qquad\square$

In the following, we use Lemma 5 to prove the generalization bounds. Specifically, we provide the proofs for the envy-free generalization bound from Eq. (8), the max-min generalization bound from Eq. (9), and the unrestricted generalization bound from Eq. (7).

**Proof of Eq. (8):**

*Proof.* It follows that

$$\sup_{\pi \in \Pi}\sup_{s,s' \in \mathcal{S}}\left|\hat{V}^{\mathrm{m}}(\pi) - \lambda\left|\hat{V}_s^{\mathrm{m}}(\pi) - \hat{V}_{s'}^{\mathrm{m}}(\pi)\right| - V(\pi) + \lambda\left|V_s(\pi) - V_{s'}(\pi)\right|\right| \qquad (44)$$

$$\leq \sup_{\pi \in \Pi}\left|\hat{V}^{\mathrm{m}}(\pi) - V(\pi)\right| + \lambda\sup_{\pi \in \Pi}\sup_{s,s' \in \mathcal{S}}\left|\left|\hat{V}_s^{\mathrm{m}}(\pi) - \hat{V}_{s'}^{\mathrm{m}}(\pi)\right| - \left|V_s(\pi) - V_{s'}(\pi)\right|\right| \qquad (45)$$

$$\leq \sup_{\pi \in \Pi}\left|\hat{V}^{\mathrm{m}}(\pi) - V(\pi)\right| + 2\lambda\sup_{\pi \in \Pi}\sup_{s \in \mathcal{S}}\left|\hat{V}_s^{\mathrm{m}}(\pi) - V_s(\pi)\right|. \qquad (46)$$

Hence, with probability of at least $1 - p_1 - p_2$, we yield

$$\sup_{\pi \in \Pi}\sup_{s,s' \in \mathcal{S}}\left|\hat{V}^{\mathrm{m}}(\pi) - \lambda\left|\hat{V}_s^{\mathrm{m}}(\pi) - \hat{V}_{s'}^{\mathrm{m}}(\pi)\right| - V(\pi) + \lambda\left|V_s(\pi) - V_{s'}(\pi)\right|\right| \qquad (47)$$

$$\leq 2CK_{\mathrm{m}}\frac{2+\nu}{\nu}\left(R_n(\Pi) + \sqrt{\frac{8\log\frac{4|\mathcal{S}|}{p_1}}{n}} + \frac{2}{(2+\nu)\sqrt{n}}\left(\frac{\ell(n, p_2)}{\nu - \frac{1}{\sqrt{n}}\ell(n, p_2)}\right)\right) \qquad (48)$$

using Lemma 5. The theorem follows from

$$\hat{V}_\lambda^{\mathrm{m}}(\pi) \leq V_\lambda(\pi) + \sup_{\pi \in \Pi}\sup_{s,s' \in \mathcal{S}}\left|\hat{V}^{\mathrm{m}}(\pi) - \lambda\left|\hat{V}_s^{\mathrm{m}}(\pi) - \hat{V}_{s'}^{\mathrm{m}}(\pi)\right| - V(\pi) + \lambda\left|V_s(\pi) - V_{s'}(\pi)\right|\right|.$$

(49)
$\square$

**Proof of Eq. (9):**

*Proof.* The triangle inequality implies that

$$\hat{V}_s^{\mathrm{m}}(\pi) \leq V_s(\pi) + \sup_{\pi \in \Pi}\left|\hat{V}_s^{\mathrm{m}}(\pi) - V_s(\pi)\right|. \qquad (50)$$

Hence, Lemma 5 yields with probability of at least $1 - p_1 - p_2$ for all $s \in \mathcal{S}, \pi \in \Pi$:

$$V_s(\pi) \geq \hat{V}_s^{\mathrm{m}}(\pi) - \frac{2CK_{\mathrm{m}}}{\nu}\left(R_n(\Pi) + \sqrt{\frac{8\log\frac{2|\mathcal{S}|}{p_1}}{n}} + \frac{1}{\sqrt{n}}\left(\frac{\ell(n, p_2)}{\nu - \frac{1}{\sqrt{n}}\ell(n, p_2)}\right)\right). \qquad (51)$$

The result follows by applying the minimum over $s$ on both sides. $\qquad\square$

**Proof of Eq.** (7):

*Proof.* It follows that

$$\hat{V}^{\mathrm{m}}(\pi) \leq V(\pi) + \sup_{\pi \in \Pi} \left| \hat{V}^{\mathrm{m}}(\pi) - V(\pi) \right|. \tag{52}$$

With the same proof as in Lemma 5, we can show, with probability of at least $1 - p$, that

$$\sup_{\pi \in \Pi} \left| \hat{V}^{\mathrm{m}}(\pi) - V(\pi) \right| \leq 2CK_{\mathrm{m}} \left( R_n(\Pi) + \sqrt{\frac{8 \log \frac{2}{p}}{n}} \right). \tag{53}$$

$\square$

## C  TOY EXAMPLE TO DIFFERENTIATE FAIRNESS NOTIONS

In the following, we provide a toy example based on which we discuss the differences between the above fairness notions. For this, we consider algorithmic decision-making in credit lending where applications for student loans are evaluated. We consider two covariates for students, namely their gender and average grade (GPA) given by $Gender \in \{\text{Female}, \text{Male}\}$ and $GPA \in \{\text{Low}, \text{High}\}$. We consider $Gender$ as the sensitive attribute $S$. The outcome $Y$ is the expected change in salary, that is, whether it increases $(= 1)$, remains the same $(= 0)$, or decreases $(= -1)$ as a result of the study program. For the purpose of our toy example, we make further assumptions regarding the distribution of covariates and expected outcomes. To this end, Table 3 reports the probability of observing an individual from each sub-population (column 3), the outcome when a student receives the loan $(\mu_1)$, and the outcome when a student does not receive the loan $(\mu_0)$. Then, the overall effect of the action (i.e., the student loan) is given by $\mu_1 - \mu_0$. As can be seen, the action of receiving a loan benefits males with a high GPA while it has a negative effect for all other sub-groups.

In Table 3, we report the policy value under different decision policies. (Details for calculating the policy values in our toy example are in Appendix C). First, we report the optimal unrestricted policy $(\pi^{\mathrm{u}})$. This policy gives student loans only to males with high GPA but not to any other student. The reason is that the sub-population of males with high GPA is the only one with a positive effect (i.e., $\mu_1 - \mu_0 = 1$). Second, we report an optimal policy under action fairness $(\pi^{\mathrm{af}})$. It chooses the same action for both males and females with high (low) GPA. Hence, the action taken by $\pi^{\mathrm{af}}$ does not depend on gender and thus fulfills action fairness. Third, envy-free fairness $(\pi^{\alpha})$ and max-min fairness $(\pi^{\mathrm{mm}})$ assign loans only to males with a high GPA. In particular, the max-min policy coincides with the optimal unrestricted policy, as implied by Lemma 1.

We further consider policies for envy-free fairness and max-min fairness that are combined with action fairness, so that always both action fairness and value fairness are satisfied (columns 10 and 11). Here, the policies assign actions to males and females with high GPA in order to fulfill action fairness. In addition, both policies assign actions only to a fraction of the overall population. This is seen by the fact that the policy outputs are $\frac{\alpha+1}{3}$ and $\frac{1}{3}$, respectively, and thus below 1. We further note that some of the policies can coincide as stipulated in Lemma 2. For $\alpha = 0$, the policy combining action fairness and envy-free fairness is identical to the policy combing action fairness and max-min fairness. For $\alpha = 2$, the policy combining action fairness and envy-free fairness is identical to the policy for action fairness $(\pi^{\mathrm{af}})$.

Table 3: Toy example comparing the different fairness notions for off-policy learning.

| Data | | | Expected outcome | | Policies | | | | Combined policies (with action fairness) | |
|---|---|---|---|---|---|---|---|---|---|---|
| Gender | GPA | Probability | $\mu_1$ | $\mu_0$ | $\pi^{\mathrm{u}}$ | $\pi^{\mathrm{af}}$ | $\pi^{\alpha}$ | $\pi^{\mathrm{mm}}$ | $\pi^{\alpha}$ | $\pi^{\mathrm{mm}}$ |
| Female | Low | 0.1 | 0 | 1 | 0 | 0 | 0 | 0 | 0 | 0 |
| Male | Low | 0.4 | 0 | 1 | 0 | 0 | 0 | 0 | 0 | 0 |
| Female | High | 0.1 | −1 | 1 | 0 | 1 | 0 | 0 | $\frac{\alpha+1}{3}$ | $\frac{1}{3}$ |
| Male | High | 0.4 | 1 | 0 | 1 | 1 | 1 | 1 | $\frac{\alpha+1}{3}$ | $\frac{1}{3}$ |

*Legend*: $\pi^{\mathrm{u}}$: optimal unrestricted; $\pi^{\mathrm{af}}$: action fairness; $\pi^{\alpha}$: envy-free fairness; $\pi^{\mathrm{mm}}$: max-min fairness

**Derivations:** We denote the levels of gender with F (female) and M (male), and the levels of GPA with L (low) and H (high). We first calculate the conditional policy value $V_{\mathrm{F}}$ for females

$$V_{\mathrm{F}}(\pi) = \pi(\mathrm{F}, \mathrm{L})\mu_1(\mathrm{F}, \mathrm{L}) + (1 - \pi(\mathrm{F}, \mathrm{L}))\mu_0(\mathrm{F}, \mathrm{L}) + \pi(\mathrm{F}, \mathrm{H})\mu_1(\mathrm{F}, \mathrm{H}) + (1 - \pi(\mathrm{F}, \mathrm{H}))\mu_0(\mathrm{F}, \mathrm{H}) \tag{54}$$

$$= (1 - \pi(\mathrm{F}, \mathrm{L})) - \pi(\mathrm{F}, \mathrm{H}) + (1 - \pi(\mathrm{F}, \mathrm{H})) \tag{55}$$

$$= 2 - \pi(\mathrm{F}, \mathrm{L}) - 2\pi(\mathrm{F}, \mathrm{H}) \tag{56}$$

and $V_{\mathrm{M}}$ for males

$$V_{\mathrm{M}}(\pi) = \pi(\mathrm{M}, \mathrm{L})\mu_1(\mathrm{M}, \mathrm{L}) + (1 - \pi(\mathrm{M}, \mathrm{L}))\mu_0(\mathrm{M}, \mathrm{L}) + \pi(\mathrm{M}, \mathrm{H})\mu_1(\mathrm{M}, \mathrm{H}) + (1 - \pi(\mathrm{M}, \mathrm{H}))\mu_0(\mathrm{M}, \mathrm{H}) \tag{57}$$

$$= 1 - \pi(\mathrm{M}, \mathrm{L}) + \pi(\mathrm{M}, \mathrm{H}). \tag{58}$$

The overall policy value is

$$V(\pi) = 0.2V_{\mathrm{F}}(\pi) + 0.8V_M(\pi) \qquad (59)$$
$$= 1.2 + 0.8\pi(\mathrm{M},\mathrm{H}) - 0.8\pi(\mathrm{M},\mathrm{L}) - 0.2\pi(\mathrm{F},\mathrm{L}) - 0.4\pi(\mathrm{F},\mathrm{H}). \qquad (60)$$

Hence, the optimal unrestricted policy is $\pi^{\mathrm{u}}(\mathrm{M},\mathrm{H}) = 1$, $\pi^{\mathrm{u}}(\mathrm{M},\mathrm{L}) = 0$, $\pi^{\mathrm{u}}(\mathrm{F},\mathrm{H}) = 0$, and $\pi^{\mathrm{u}}(\mathrm{F},\mathrm{L}) = 0$.

The difference of the conditional policy value is

$$\Delta(\pi) = |V_{\mathrm{F}}(\pi) - V_{\mathrm{M}}(\pi)| = |1 - \pi(\mathrm{F},\mathrm{L}) - 2\pi(\mathrm{F},\mathrm{H}) - \pi(\mathrm{M},\mathrm{H}) + \pi(\mathrm{M},\mathrm{L})|. \qquad (61)$$

It holds that $\Delta(\pi^{\mathrm{u}}) = 0$ which implies that the policy $\pi^{\alpha}$ with $\alpha$-envy-free fairness coincides with $\pi^{\mathrm{u}}$.

For the optimal policy $\pi^{\mathrm{af}}$ with action fairness, the policy value simplifies to

$$V(\pi^{\mathrm{af}}) = 1.2 + 0.8\pi^{\mathrm{af}}(\mathrm{H}) - 0.8\pi^{\mathrm{af}}(\mathrm{L}) - 0.2\pi^{\mathrm{af}}(\mathrm{L}) - 0.4\pi^{\mathrm{af}}(\mathrm{H}) \qquad (62)$$
$$= 1.2 + 0.4\pi^{\mathrm{af}}(\mathrm{H}) - \pi^{\mathrm{af}}(\mathrm{L}), \qquad (63)$$

which means that $\pi^{\mathrm{af}}(\mathrm{L}) = 0$ and $\pi^{\mathrm{af}}(\mathrm{H}) = 1$. For the policy $\pi^{\mathrm{af}+\alpha}$ with both action fairness and envy-free fairness, we obtain

$$\Delta(\pi^{\mathrm{af}+\alpha}) = |1 - 3\pi^{\mathrm{af}+\alpha}(\mathrm{H})| \le \alpha, \qquad (64)$$

which yields $\pi^{\mathrm{af}+\alpha}(\mathrm{L}) = 0$ and $\pi^{\mathrm{af}+\alpha}(\mathrm{H}) = \frac{\alpha+1}{3}$. Finally, the policy $\pi^{\mathrm{mm}}$ with max-min fairness maximizes

$$\min\{V_{\mathrm{F}}(\pi^{\mathrm{mm}}), V_{\mathrm{M}}(\pi^{\mathrm{mm}})\} = \min\{2 - \pi^{\mathrm{mm}}(\mathrm{L}) - 2\pi^{\mathrm{mm}}(\mathrm{H}), \ 1 - \pi^{\mathrm{mm}}(\mathrm{L}) + \pi^{\mathrm{mm}}(\mathrm{H})\}, \quad (65)$$

which implies $\pi^{\mathrm{mm}}(\mathrm{L}) = 0$ and $\pi^{\mathrm{mm}}(\mathrm{H}) = 1/3$.

# D DISCUSSION OF THE ASSUMPTIONS IN LEMMA 2

In this section, we provide additional details regarding the assumptions in Lemma 2. In essence, Lemma 2 holds if the max-min solution $(\pi^{\mathrm{mm}}, s^*)$ outputs stochastic actions in an area $V$ of the covariate space where the policy value could be improved by choosing a deterministic action. In the toy example from the previous section, this is the case, and, hence, the max-min and the envy-free policy with $\alpha = 0$ coincide. In the following, we study a second toy example where $\pi^{\mathrm{mm}}$ does not coincide with any envy-free policy $\pi^{\mathrm{af}+\alpha}$.

## D.1 TOY EXAMPLE

The same data from Table 3 is shown in Table 4 with different ITEs. Now, the action benefits all groups, but males have larger expected outcomes than females. Furthermore, old males receive a larger benefit from the action than all other groups. Hence, even though the action benefits all groups, the difference in policy values for males and females will increase by performing actions for old male patients. The max-min policy $\pi^{\mathrm{mm}}$ simply recommends action to everyone, as it aims to maximize the policy value $V_{\mathrm{Female}}(\pi^{\mathrm{mm}})$ for females (worst-case). In contrast, the $\pi^{\mathrm{af}+\alpha}$ restricts action on older patients in order to decrease the disparity of conditional policy values $V_{\mathrm{Female}}(\pi^{\mathrm{af}+\alpha})$ and $V_{\mathrm{Male}}(\pi^{\mathrm{af}+\alpha})$ (envy-free).

Table 4: Toy example comparing the different fairness notions for off-policy learning.

| Data | | | Expected outcome | | Policies | | | | Combined policies (with action fairness) | |
|---|---|---|---|---|---|---|---|---|---|---|
| Gender | GPA | Probability | $\mu_1$ | $\mu_0$ | $\pi^{\mathrm{u}}$ | $\pi^{\mathrm{af}}$ | $\pi^{\alpha}$ | $\pi^{\mathrm{mm}}$ | $\pi^{\alpha}$ | $\pi^{\mathrm{mm}}$ |
| Female | Low | 0.1 | 0 | $-1$ | 1 | 1 | 1 | 1 | 1 | 1 |
| Male | Low | 0.4 | 1 | 0 | 1 | 1 | $\frac{\alpha}{3}$ | 1 | 1 | 1 |
| Female | High | 0.1 | 0 | $-1$ | 1 | 1 | 1 | 1 | $\alpha - 2$ | 1 |
| Male | High | 0.4 | 2 | 0 | 1 | 1 | $\frac{\alpha}{3}$ | 1 | $\alpha - 2$ | 1 |

*Legend*: $\pi^{\mathrm{u}}$: optimal unrestricted; $\pi^{\mathrm{af}}$: action fairness; $\pi^{\alpha}$: envy-free fairness; $\pi^{\mathrm{mm}}$: max-min fairness

## D.2 DERIVATION OF TOY EXAMPLE

We proceed as in the example from our main paper (see Appendix C for details) and calculate the conditional policy values

$$V_{\mathrm{F}}(\pi) = -(1 - \pi(\mathrm{F}, \mathrm{L})) - (1 - \pi(\mathrm{F}, \mathrm{H})) \tag{66}$$
$$= \pi(\mathrm{F}, \mathrm{L}) + \pi(\mathrm{F}, \mathrm{H}) - 2 \tag{67}$$

and

$$V_{\mathrm{M}}(\pi) = \pi(\mathrm{M}, \mathrm{L}) + 2\pi(\mathrm{M}, \mathrm{H}). \tag{68}$$

The overall policy value is

$$V(\pi) = 0.2 V_{\mathrm{F}}(\pi) + 0.8 V_{\mathrm{M}}(\pi) \tag{69}$$
$$= 1.6\pi(\mathrm{M}, \mathrm{H}) + 0.8\pi(\mathrm{M}, \mathrm{L}) + 0.2\pi(\mathrm{F}, \mathrm{L}) + 0.2\pi(\mathrm{F}, \mathrm{H}) - 0.4 \tag{70}$$

We immediately obtain the optimal unrestricted policy is $\pi^{\mathrm{u}} \equiv \pi^{\mathrm{af}} \equiv \pi^{\mathrm{mm}} \equiv 1$ because all policy terms are positive. To obtain policies $\pi^{\mathrm{af}+\alpha}$ and $\pi^{\alpha}$ with envy-free fairness, we write the constraints as

$$\Delta(\pi^{\alpha}) = |2 + \pi^{\alpha}(\mathrm{M}, \mathrm{L}) + 2\pi^{\alpha}(\mathrm{M}, \mathrm{H}) - \pi^{\alpha}(\mathrm{F}, \mathrm{L}) - \pi^{\alpha}(\mathrm{F}, \mathrm{H})| \le \alpha \tag{71}$$

and

$$\Delta(\pi^{\mathrm{af}+\alpha}) = 2 + \pi^{\mathrm{af}+\alpha}(\mathrm{H}) \le \alpha, \tag{72}$$

where $\pi^{\mathrm{af}+\alpha}$ again denotes the policy that fulfills both action fairness and envy-free fairness. Eq. (72) implies $\pi^{\mathrm{af}+\alpha}(\mathrm{H}) = \alpha - 2$ (for $\alpha \ge 2$) and $\pi^{\mathrm{af}+\alpha}(\mathrm{L}) = 1$. Eq. (71) yields a linear constrained optimization problem with solution $\pi^{\alpha}(\mathrm{F}, \mathrm{L}) = \pi^{\alpha}(\mathrm{F}, \mathrm{H}) = 1$ and $\pi^{\alpha}(\mathrm{M}, \mathrm{L}) = \pi^{\alpha}(\mathrm{M}, \mathrm{H}) = \alpha/3$.

# E LEARNING ALGORITHM FOR FAIRPOL

Algorithm 1 provides the learning algorithm for FairPol. The algorithm consists of two consecutive steps, namely the fair representation learning step and the policy learning step. For the policy learning step, the specification of the policy loss is needed, namely one of no value, envy-free fairness, and max-min fairness: $\mathcal{L}_\pi^{\mathrm{m}}(\cdot) \in \{\hat{V}^{\mathrm{m}}(\cdot), \hat{V}_\lambda^{\mathrm{m}}(\cdot), and \min_{s \in S} \hat{V}_s^{\mathrm{m}}(\cdot)\}$.

---

**Algorithm 1** Learning algorithm for FairPol

---

**Input:** hyperparameters of representation networks (number of epochs $n_r$, learning rate $\eta_r$, action fairness parameter $\gamma$), hyperparameters of policy network (number of epochs $n_p$, learning rate $\eta_p$, policy loss $\mathcal{L}_\pi^{\mathrm{m}}(\cdot)$)

Initialize $\theta_Y^{(0)}, \theta_\Phi^{(0)}, \theta_S^{(0)} \sim$ Kaiming-Uniform       $\triangleright$ Step 1. Fitting the representation networks

**for** $k = 1$ **to** $n_r$ **do**

  Forward pass of the base representation, outcome prediction, and sensitive attribute networks with $\theta_Y^{(k-1)}, \theta_\Phi^{(k-1)}, \theta_S^{(k-1)}$

  $\theta_Y^{(k)} \leftarrow \theta_Y^{(k-1)} - \eta_r \nabla_{\theta_Y} \left[ \mathcal{L}_Y(\theta_\Phi^{(k-1)}, \theta_Y^{(k-1)}) \right]$

  $\theta_\Phi^{(k)} \leftarrow \theta_\Phi^{(k-1)} - \eta_r \nabla_{\theta_\Phi} \left[ \mathcal{L}_Y(\theta_\Phi^{(k-1)}, \theta_Y^{(k-1)}) + \gamma \mathcal{L}_{\mathrm{conf}}(\theta_\Phi^{(k-1)}, \theta_S^{(k-1)}) \right]$

  Forward pass of the base representation and sensitive attribute networks with $\theta_\Phi^{(k)}, \theta_S^{(k-1)}$

  $\theta_S^{(k)} \leftarrow \theta_S^{(k-1)} - \eta_r \nabla_{\theta_S} \left[ \gamma \mathcal{L}_S(\theta_\Phi^{(k)}, \theta_S^{(k-1)}) \right]$

**end for**

$\hat{\theta}_\Phi \leftarrow \theta_\Phi^{(n_r)}$

Initialize $\theta^{(0)} \sim$ Kaiming-Uniform       $\triangleright$ Step 2. Fitting the policy network

**for** $k = 1$ **to** $n_p$ **do**

  Forward pass of the policy network with $\theta^{(k-1)}$

  $\theta^{(k)} \leftarrow \theta^{(k-1)} - \eta_p \nabla_\theta \left[ \mathcal{L}_\pi^{\mathrm{m}}(\pi_{\theta^{(k-1)}}(\hat{\Phi}(X))) \right]$

**end for**

---

# F  HYPERPARAMETER TUNING

We followed best practices in causal machine learning (e.g., Bica et al., 2021; Curth & van der Schaar, 2021) and performed extensive hyperparameter tuning for FairPol. We split the data into a training set (80%) and a validation set (10%). We then performed 30 random grid search iterations and chose the set of parameters that minimized the respective training loss on the validation set. In particular, the tuning procedure was the same for all baselines, which ensures that the performance differences reported in Section 6 are not due to larger flexibility but are due to the different methods themselves.

We performed hyperparameter tuning for all neural networks in FairPol, i.e., the different representation networks and the policy network. For the real-world data, we also used TARNet (Shalit et al., 2017) in order to estimate the nuisance parameters. We first performed hyperparameter tuning for TARNet and for the representation networks, before tuning the policy neural networks by using the input from the tuned neural networks. The tuning ranges for the hyperparameter are shown in Table 5 (simulated data) and Table 6 (real-world data).

Table 5: Hyperparameter tuning ranges (simulated data).

| NEURAL NETWORK | HYPERPARAMETER | TUNING RANGE |
|---|---|---|
| All neural networks | Dropout probability | 0, 0.1, 0.2 |
| | Batch size | 32, 64, 128 |
| | Epochs | 400 |
| Representation networks | Learning rate | 0.0001, 0.0005, 0.001, 0.005 |
| | Hidden layer / representation size | 2, 5, 10 |
| | Weight decay | 0, 0.001 |
| Policy network | Learning rate | 0.00005, 0.0001, 0.0005, 0.001 |
| | Hidden layer size | 5, 10, 15, 20 |
| | Weight decay | 0 |

Table 6: Hyperparameter tuning ranges (real-world data).

| NEURAL NETWORK | HYPERPARAMETER | TUNING RANGE |
|---|---|---|
| All neural networks | Dropout probability | 0, 0.1, 0.2, 0.3 |
| | Batch size | 32, 64, 128 |
| TARNet | Learning rate | 0.0001, 0.0005, 0.001, 0.005 |
| | Hidden layer sizes | 5, 10, 20, 30 |
| | Weight decay | 0 |
| | Epochs | 200 |
| Representation networks | Learning rate | 0.0001, 0.0005, 0.001, 0.005 |
| | Hidden layer / representation size | 2, 5, 10 |
| | Epochs | 400 |
| | Weight decay | 0, 0.001 |
| Policy network | Learning rate | 0.00005, 0.0001, 0.0005, 0.001 |
| | Hidden layer size | 5, 10, 15, 20 |
| | Epochs | 300 |
| | Weight decay | 0 |

The tables include both the hyperparameter ranges shared across all neural networks and the network-specific hyperparameters. For reproducibility purposes, we report the selected hyperparameters as *.yaml* files.[5]

---

[5]Codes are in the supplementary materials and at `https://anonymous.4open.science/r/FairPol-402C`.

# G DETAILS REGARDING SIMULATED DATA

Here, we provide details regarding our synthetic data generation. We consider a decision problem from credit lending where loans are approved based on covariates of the customers, yet where algorithmic decision-making must not discriminate by gender. To this end, we denote the sensitive attribute by $S \in \{0, 1\}$ and generate data as follows. We simulate two covariates $X_\mathrm{u} \in \mathbb{R}$ and $X_s \in \mathbb{R}$ via

$$S \sim \mathrm{Bernoulli}(p_s), \qquad X_u \sim \mathcal{U}[-1, 1], \qquad \text{and} \qquad X_s \mid S = s \sim \mathcal{U}[s-1, s], \qquad (73)$$

where $\mathcal{U}[-1, 1]$ is the uniform distribution over the interval $[-1, 1]$. Thus, $X_u$ is independent of $S$, while $X_s$ is correlated with $S$. In practice, $X_u$ can be, e.g., a credit score (which gives the probability of repaying a loan yet which is independent of gender), while $X_s$ can be, e.g., income (which is often correlated with gender). We further generate actions (decisions on whether a loan was approved or not) via

$$A \mid X_u = x_u, X_s = x_s, S = s \sim \mathrm{Bernoulli}(p) \quad \text{with} \quad p = \sigma(\sin(2x_u) + \sin(2X_s) + \sin(2s)), \qquad (74)$$

where $\sigma(\cdot)$ denotes the sigmoid function. Finally, we generate outcomes

$$Y = \mathbb{1}\{A = 1\}\left(\mathbb{1}\{X_u < 0.5\}\sin(4X_s - 2) + \mathbb{1}\{X_u > 0.5\}(0.6\,S - 0.3)\right) + \epsilon, \qquad (75)$$

where $\epsilon \sim \mathcal{N}(0, 0.1)$. In our example, the outcomes could correspond to the profit for the lending institution. We sample a dataset of $n = 3000$ sample from the data generating process and split the data into a training set (80%) and a test set (20%).

# H DETAILS REGARDING REAL-WORLD DATA

In our experiment with real-world data, we use data from the Oregon health insurance experiment (OHIE)[6] (Finkelstein et al., 2012). The OHIE was conducted as a large-scale experiment among public health to assess the effect of health insurance on several outcomes such as health or economic status. In 2008, a lottery draw offered low-income, uninsured adults in Oregon participation in a Medicaid program, providing health insurance. Chosen individuals were offered free health insurance. After a period of 12 months, a survey was conducted to evaluate several outcomes of the participants.

In our analysis, the decision to sign up for the Medicaid program is the action $A$, and the overall out-of-pocket cost for medical care within the last 6 months is the outcome $Y$. The sensitive covariate $S$ we consider is gender. Furthermore, we include the following covariates $X$: age, the number of people the individual signed up with, the week the individual signed up, the number of emergency visits before the experiment, and language. We extract $n = 24,646$ observations from the OHIE data and plot the histograms of all variables in Fig. 5.

We split the data randomly into a train (0.7%), validation (0.1%), and test set (0.2%) and perform hyperparameter tuning using the validation set. The evaluation metrics are then computed using the test set. We estimate the nuisance parameters using a TARNet (Shalit et al., 2017), for which we perform hyperparameter tuning according to Appendix F. Then, we used the estimators proposed in Sec. 5.2 to estimate (conditional) policy values using the estimated nuisance parameters).

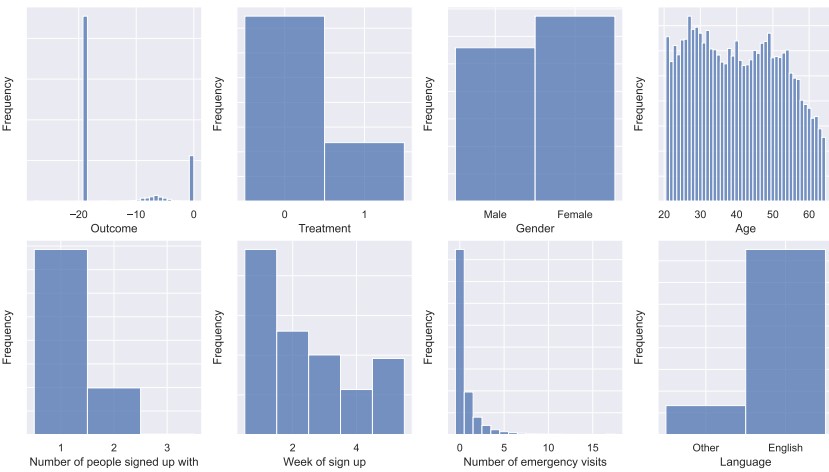

Figure 5: Histograms of marginal distributions (real-world data).

[6]The dataset is available here: https://www.nber.org/programs-projects/projects-and-centers/oregon-health-insurance-experiment

# I DISCUSSION

Unfair decisions can have detrimental consequences for individuals because of which ethical and legal frameworks require that algorithmic decision-making must ensure fairness (Barocas & Selbst, 2016; Kleinberg et al., 2019). Hence, potential applications benefiting from fairness for algorithmic decision-making are vast and include healthcare, lending, and criminal justice, among many others (De-Arteaga et al., 2022). For instance, in the U.S., the Equal Credit Opportunity Act mandates that lending decisions are fair for individuals of different gender, race, and other sensitive groups, while the Fair Housing Act enforces similar principles for housing rentals and purchases. As such algorithmic decision-making must avoid discrimination of individuals and thus generate decisions that are regarded as fair.

We addressed the problem of fairness in algorithmic decision-making by learning fair policies from observational data. Our framework has three main benefits that make it appealing for use in practice: (1) Our framework is directly applicable even in settings where observational data ingrain historical discrimination. Despite such historical discrimination, our framework can still obtain a fair policy. This is relevant for practice as there is a growing awareness that many data sources are biased and that it is often challenging or infeasible to remove bias in historical data (Corbett-Davies et al., 2023). (2) Our framework comes with a scalable machine learning instantiation based on a custom neural network (FairPol). Hence, practitioners can effectively generate fair policies from high-dimensional and non-linear observational data. (3) Our framework is flexible in the sense that it supports different fairness notions. Practitioners can thus adapt our framework to the underlying fairness goals as well as the legal and ethical contexts and thus choose a suitable fairness notion. Together, our framework fulfills crucial fairness demands in many applications from practice (e.g., automated hiring, credit lending, and ad targeting).

Our work contributes to the literature in several ways. First, our work connects to off-policy learning (e.g., Kallus, 2018; Athey & Wager, 2021). While there is a growing body of literature that uses off-policy learning for managerial decision-making such as pricing and ad targeting (e.g., Smith et al., 2023; Yoganarasimhan et al., 2022; Yang et al., 2023), we add by offering a new framework with fairness guarantees. In particular, our work fills an important gap in the literature in that we are able to learn fair policies from discriminatory observational data. Second, there is extensive literature on algorithmic fairness that focuses on machine learning predictions (e.g., Hardt et al., 2016; Kusner et al., 2017; Nabi & Shpitser, 2018), whereas we contribute to algorithmic fairness for decision-making from observational data, specifically, off-policy learning. Third, fairness notions such as envy-free fairness or max-min fairness have been used in traditional, utility-based decision models such as those from resource allocation and pricing (e.g., Bertsimas et al., 2011; Kallus & Zhou, 2021; Cohen et al., 2022). We build upon these fairness notions but integrate them into off-policy learning.

**Conclusion:** In this paper, we proposed a novel framework for fair off-policy learning from observational data. For this, we introduced fairness notions tailored to off-policy learning, then developed a flexible instantiation of our framework using machine learning (FairPol), and finally provided theoretical guarantees in form of generalization bounds. Our framework is widely applicable to algorithmic decision-making in practice where fairness must be ensured.

# J PRACTICAL RECOMMENDATIONS FOR USING OUR FAIRNESS CRITERIA

We discuss the applicability of our three proposed fairness criteria (action fairness, envy-free fairness, and max-min fairness) and when practitioners should choose which. Table 7 lists the advantages and disadvantages of the different fairness criteria as well as examples of real-world use cases in which practitioners may consider enforcing these.

| Fairness metric | Definition | Use cases | Advantages/ disadvantages |
|---|---|---|---|
| *Action fairness* | A policy fulfills action fairness if the recommended action is independent of the sensitive attribute. $\Rightarrow$ Definition 1 | Hiring, credit lending, where decisions cannot directly discriminate against/depend on sensitive attributes. | ⚠ Focuses on the underlying mechanism for assigning treatments, not outcomes/utility from treatments. ⊕ Ensures equal treatment across sensitive attributes. ⊕ Consistent with many regulatory frameworks (e.g., US Fair Lending Laws, US Fair Housing Act, anti-discrimination laws in the EU). ⊕ Typically suitable when non-treatment does not lead to immediate loss or harm. ⊕ Often suitable for selection tasks where items out of a large pool should be chosen. ⊖ Does not consider discrepancies in policy values between the sensitive groups. |
| *Envy-free fairness* | A policy fulfills envy-free fairness if the conditional policy values between sub-populations do not differ more than a predefined level. $\Rightarrow$ Definition 2 | Resource allocation, scholarships, where disparities in utility between groups are controlled. | ⚠ Focuses on outcomes/utility of an individual relative to others. ⊕ Allows control over utility disparities. ⊕ Typically suitable for distributive tasks where a given pool of $N$ items should be allocated. ⊕ Grounded in ethics (Rawlsian justice). ⊖ May require careful calibration of the fairness level. ⊖ May lead to policies that are not Pareto optimal, i.e., the policy value of one sensitive group may be improved without reducing the policy value for the opposite group. |
| *Max-min fairness* | A policy fulfills max-min fairness if it maximizes the worst-case policy value for the sensitive attributes. $\Rightarrow$ Definition 3 | Situations where the worst-case outcome is to be optimized, e.g., emergency services allocation. | ⚠ Focuses on the worst outcome/utility across a group. ⚠ When combined with action fairness, can be a special case of envy-free fairness. ⊕ Focuses on improving the worst-off group. ⊕ Oftentimes helpful for decision-makers that "do not want to harm". ⊕ Typically suitable for distributive tasks where a given pool of $N$ items should be allocated. ⊖ Could lead to suboptimal outcomes for non-worst-off groups. |

Table 7: Guidance for choosing our fairness metrics for policy learning in practice.

# K SYNTHETIC EXPERIMENTS WITH ADDITIONAL REPRESENTATION LEARNING BASELINES

Here, we compare our approach for fair representation learning (=adversarial domain confusion loss) against potential benchmarks. In principle, our approach for fair representation learning (=adversarial domain confusion loss) can be replaced by any other approach that aims at enforcing independence from the sensitive attribute. In the following, we consider two common baselines from the literature: (i) adversarial learning using gradient reversal (Ganin & Lempitsky, 2015); and (ii) regularization using a Wasserstein distance Shalit et al. (2017). Both are as follows:

**(i) Adversarial learning using gradient reversal:** Here, we consider an adversarial approach that reduces the dependence between the representation and sensitive attribute via gradient reversal. That is, we defined the loss

$$\mathcal{L}_\gamma(\theta_\Phi, \theta_S, \theta_Y) = \mathcal{L}_Y(\theta_\Phi, \theta_Y) - \gamma \mathcal{L}_S(\theta_\Phi, \theta_S) \tag{76}$$

and solve the adversarial problem

$$\hat{\theta}_\Phi, \hat{\theta}_Y = \arg\min_{\theta_\Phi, \theta_Y} \mathcal{L}_\gamma(\theta_\Phi, \hat{\theta}_S, \theta_Y); \qquad \hat{\theta}_S = \arg\max_{\theta_S} \mathcal{L}_\gamma(\hat{\theta}_\Phi, \theta_S, \hat{\theta}_Y); \tag{77}$$

where $\gamma$ is a parameter that weights the different parts in the loss function. For further details, we refer to Ganin & Lempitsky (2015) or Bica et al. (2020).

**(ii) Regularization using Wasserstein distance:** Here, we consider a regularization approach similar to Shalit et al. (2017) that solves

$$\hat{\theta}_\Phi, \hat{\theta}_Y = \arg\min_{\theta_\Phi, \theta_Y} \mathcal{L}_Y(\theta_\Phi, \theta_Y) + \gamma \mathcal{W}_p(\{\Phi_{\theta_\Phi}(X_i)\}_{S=1}, \{\Phi_{\theta_\Phi}(X_i)\}_{S=0}), \tag{78}$$

where $mathcalW_p$ denotes the $p$-Wasserstein distance between the empirical distributions $\{\Phi_{\theta_\Phi}(X_i)\}_{S=1}$ and $\{\Phi_{\theta_\Phi}(X_i)\}_{S=0}$, and $\gamma$ is a parameter that weights the strength of the regularization.

**Implementation:** For both baselines, we use Step 1 of FairPol as the base architecture (see Sec. 5.1 for details) and use the same neural network-specific hyperparameters (see Appendix F). We choose $p = 2$ for the Wasserstein distance.

**Experiments:** We then train FairPol without value fairness and only action fairness, where the action fairness is enforced via (1) adversarial domain confusion, (2) adversarial gradient reversal, and (3) regularization using Wasserstein distance, respectively. We repeat the training of FairPol for different values of $\gamma$, and plot the corresponding action fairness and achieved policy value in Fig. 6. The adversarial domain confusion loss performs consistently best over different levels of action fairness. This is also consistent with prior literature (Melnychuk et al., 2022).

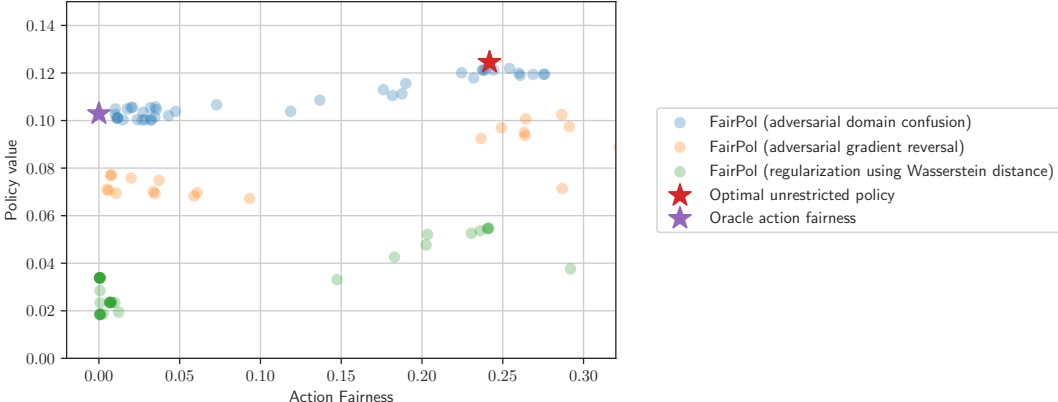

Figure 6: Comparing different representation learning methods in Step 1 for FairPol to enforce action fairness.

## L  SYNTHETIC EXPERIMENTS WITH ESTIMATED NUISANCE PARAMETERS

In this section, we repeat the experiments from Table 1in Sec. 6 but where we estimate nuisance parameters $\mu_j(X, S) = \mathbb{E}[Y \mid X, S, A = j], j \in \{0, 1\}$ and $\pi_b(X, S) = \mathbb{P}(A = 1 \mid X, S)$ from the observational data. To do so, we use a TARNet (Shalit et al., 2017) for estimation and refer to Appendix H for details. The results are shown in Table 8. In particular, our results from Table 1 are robust with respect to estimation errors in nuisance parameters. In sum, this demonstrates the effectiveness of our framework when nuisance parameters are estimated from data.

Table 8: Results for simulated data with estimated nuisance parameters.

| Approach | Policy value | | | Action fairness |
|---|---|---|---|---|
| | Overall | $S = 0$ | $S = 1$ | |
| OUR FAIRPOL (ONLY ACTION FAIRNESS) | | | | |
| FairPol with $\mathrm{m} = \mathrm{DM}$ | $0.88 \pm 0.16$ | $0.34 \pm 0.34$ | $1.11 \pm 0.36$ | $0.17 \pm 0.06$ |
| FairPol with $\mathrm{m} = \mathrm{IPW}$ | $1.02 \pm 0.01$ | $0.00 \pm 0.05$ | $1.45 \pm 0.07$ | $0.20 \pm 0.04$ |
| FairPol with $\mathrm{m} = \mathrm{DR}$ | $1.02 \pm 0.01$ | $0.00 \pm 0.05$ | $1.45 \pm 0.07$ | $0.18 \pm 0.04$ |
| OUR FAIRPOL WITH ENVY-FREE FAIRNESS | | | | |
| FairPol with $\mathrm{m} = \mathrm{DM}$ | $0.85 \pm 0.15$ | $0.61 \pm 0.12$ | $0.95 \pm 0.24$ | $0.51 \pm 0.44$ |
| FairPol with $\mathrm{m} = \mathrm{IPW}$ | $0.80 \pm 0.05$ | $0.49 \pm 0.14$ | $0.94 \pm 0.11$ | $0.14 \pm 0.07$ |
| FairPol with $\mathrm{m} = \mathrm{DR}$ | $0.83 \pm 0.05$ | $0.43 \pm 0.12$ | $1.00 \pm 0.11$ | $0.22 \pm 0.07$ |
| OUR FAIRPOL WITH MAX-MIN FAIRNESS | | | | |
| FairPol with $\mathrm{m} = \mathrm{DM}$ | $0.72 \pm 0.04$ | $0.72 \pm 0.04$ | $0.73 \pm 0.04$ | $0.13 \pm 0.06$ |
| FairPol with $\mathrm{m} = \mathrm{IPW}$ | $0.73 \pm 0.03$ | $0.73 \pm 0.03$ | $0.73 \pm 0.03$ | $0.14 \pm 0.08$ |
| FairPol with $\mathrm{m} = \mathrm{DR}$ | $0.73 \pm 0.03$ | $0.73 \pm 0.03$ | $0.73 \pm 0.03$ | $0.11 \pm 0.04$ |

Reported: mean $\pm$ standard deviation ($\times 10$) on test set over 5 runs.

# M  SYNTHETIC EXPERIMENTS WITH VARYING SAMPLE SIZES

Here we repeat the experiment from Table 1 with three different samples sizes $n \in \{1000, 3000, 5000\}$. The results are shown in Table 9.

Table 9: Results for simulated data with varying sample sizes ($m = $ DR).

| Approach | Policy value | | | Action fairness |
|---|---|---|---|---|
| | Overall | $S = 0$ | $S = 1$ | |
| BASELINES | | | | |
| Optimal unrestricted policy | $1.24 \pm 0.03$ | $0.74 \pm 0.03$ | $1.46 \pm 0.06$ | $2.42 \pm 0.20$ |
| Oracle action fairness | $1.03 \pm 0.02$ | $0.01 \pm 0.07$ | $1.46 \pm 0.06$ | $0.00 \pm 0.00$ |
| OUR FAIRPOL (ONLY ACTION FAIRNESS) | | | | |
| $n = 1000$ | $0.93 \pm 0.09$ | $0.09 \pm 0.16$ | $1.28 \pm 0.04$ | $0.53 \pm 0.69$ |
| $n = 3000$ | $1.01 \pm 0.03$ | $0.02 \pm 0.05$ | $1.43 \pm 0.07$ | $0.23 \pm 0.05$ |
| $n = 3000$ | $1.01 \pm 0.03$ | $0.00 \pm 0.05$ | $1.45 \pm 0.07$ | $0.15 \pm 0.05$ |
| OUR FAIRPOL WITH ENVY-FREE FAIRNESS | | | | |
| $n = 1000$ | $0.68 \pm 0.15$ | $0.37 \pm 0.22$ | $0.81 \pm 0.19$ | $0.42 \pm 0.60$ |
| $n = 3000$ | $0.86 \pm 0.06$ | $0.34 \pm 0.17$ | $1.09 \pm 0.15$ | $0.26 \pm 0.10$ |
| $n = 5000$ | $0.81 \pm 0.03$ | $0.44 \pm 0.09$ | $0.97 \pm 0.05$ | $0.18 \pm 0.14$ |
| OUR FAIRPOL WITH MAX-MIN FAIRNESS | | | | |
| $n = 1000$ | $0.73 \pm 0.03$ | $0.73 \pm 0.03$ | $0.73 \pm 0.03$ | $0.13 \pm 0.10$ |
| $n = 3000$ | $0.73 \pm 0.03$ | $0.73 \pm 0.03$ | $0.73 \pm 0.03$ | $0.13 \pm 0.03$ |
| $n = 5000$ | $0.73 \pm 0.03$ | $0.73 \pm 0.03$ | $0.73 \pm 0.03$ | $0.18 \pm 0.10$ |

Reported: mean $\pm$ standard deviation ($\times 10$) on test set over 5 runs.

