# OpenReview forum: "Fair Off-Policy Learning from Observational Data"
_ICLR.cc/2024/Conference — Submitted to ICLR 2024_

### Official Review · Reviewer_3M28 · 2023-10-22

**Soundness:** 2 fair
**Presentation:** 2 fair
**Contribution:** 2 fair
**Rating:** 6
**Confidence:** 4

**Summary:**

This paper studies how to enforce user-side fairness in (off-)policy learning. Specifically, the paper considers allocating (i) the same action choice probability regardless user's sensitive attributes (e.g., allocating the decisions of credit lending $A \in \{0, 1\}$ regardless of their gender) and (ii) the same benefits (i.e., conditional policy value) for different user demographic groups. For this, the paper proposes the combination of the following two methods. The first one is fair representation learning, which aims to learn a feature $\Phi(X)$ which is predictive for the outcome ($Y$), but is not predictive for the sensitive feature ($S$). The second is to impose three types of fairness constraints to the policy learning objectives. The results on synthetic and real-world datasets show that the proposed method performs better than a naive baseline that myopically optimizes for the policy value without the fairness constraints.

**Strengths:**

1. **Conceptually interesting work.** Considering fairness in off-policy learning is conceptually novel and is an interesting discussion to bring to the community. The need of discussing fairness is also well-motivated.

2. **Reasonable approaches for enforcing fairness.** Learning fair representation to avoid discriminative action choices and imposing value-based constraints seems to be a reasonable approach to address fairness issues in policy learning.

3. **No restrictive assumptions of the policy class.** This paper does not constrain any assumptions on reward and the parametrization of policy. Thus, the framework is applicable to any machine learning models, as discussed in related work.

**Weaknesses:**

1. **Some formulation and mathematical descriptions should be double-checked.**
Specifically, there are some contradictions between the text description and the mathematical formulation of the *action fairness*. As the definition of action fairness should affect the theoretical analysis and empirical evaluations in the rest of the paper, this seems to be a critical concern.  Please refer to Question 1 for details.

2. **Related work seems insufficient.**
Specifically, while the paper proposes a fair representation learning method, no discussion of fair representation learning is provided in the related work. It was thus difficult to judge if the proposed approach is novel enough from the manuscript due to the lack of this discussion.

3. **No discussion on how the specific problem of off-policy learning has been resolved.** In my understanding, enforcing fairness in OPL is difficult because $V_s$  $(= \mathbb{E}_{A \sim \pi(X)}[Y | X, A, S=s])$, which is necessary for imposing the constraints on "value fairness", is not easily estimated with biased logged data. However, there is no explicit discussion on this point, and it seems that the paper just applied what has been successful in the online setting to the off-policy setting, without discussing particular challenges we will face in OPL. It is not clear what kind of OPL-related problems this paper tries to solve, and also there is no discussion about this point in the related work. It would also be beneficial to have additional ablations for this part in the experiments as well.

4. **Results of the real-world experiments do not seem promising.** The proposed approach improves "action fairness", however, it does not improve or even sometimes worsens the "value fairness" compared to the naive, unconstrained policy. Justifications are needed for this point.

5. **The clarity of writing has some room for improvement.** Aside from the above points, this paper has some additional ambiguous parts. Please refer to Questions 2 and 3.

**Questions:**

1. **Definition of action fairness**

First, the paper defines the action fairness as follows.

> a policy $\pi$ should assign the same action $A=A'$ to two individuals with the same covariates $X = X'$ but different gender $S=S'$.

However, the paper repeatedly mentions that covariates $X$ can also be a factor of unfairness even if the algorithm does not explicitly make decisions on the sensitive attributes $S$ as follows.

> in observational data, **other variables may act as proxies of for gender, and, hence, the learned policy may still lead to discrimination** due to the underlying data-generation process. (Introduction)

> Action fairness is .. It ensures that individuals who only differ with respect to their sensitive attributes (**and covariates correlated with them**) receive the same decision. (Section 4)

Given this kind of confoundings, I guess the action fairness should instead be defined as follows.

*Let $\bar{X} \subseteq X$ be a covariate that does not depend on the sensitive attribute $S$ (i.e., $\bar{X} \perp S$). Then, a policy $\pi$ should assign the same action $A=A'$ to two individuals with the same covariates $X \setminus \bar{X} = X' \setminus \bar{X'}$ but different gender $S \neq S'$ and the associated attributes $\bar{X} \neq \bar{X'}$*.

This modification will be needed because we can consider the following situation: $X_0 = f(S_0)$ and $X_{1:m}$ is drawn independent of $S_0$. In this case, the original condition can still cause discrimination because the policy can still allocate different actions depending on $X_0$ even if the policy does not explicitly depend on $S$ (i.e., "$\pi(X) \perp S$" as described in the paper).

Could you provide any justification for this point?

2. **"*Parato-optimal*" and "*general*" policies in related work**

The paper cites Viviano & Bradic, 23 as one of the most important related works and discusses the differences. However, it was not clear for me what are "*Parato-optimal*" and "*general*" policies refer to. Clarifications are needed for this point.

3. **Experimental details**

In the synthetic experiment section, I found this description:

> We provide the results for our framework across three different policy scores, namely $m \in $ {DM, IPS, DR}.

However, Table 1 reports the results of only one of them, and the section does not describe which method is used. Additionally, I could not find any additional results even in the Appendix. Could you provide some clarification on this? I also suggest reporting the number of random seeds used for the experiment.

**Some other suggestions for improvement (not really weaknesses)**
- On page 4, the paper denotes "$\mu_j(X, S) = \mathbb{E}[Y | X, S, A = j], j = \{ 0, 1 \}$, are the outcome regression functions" (of DM and DR). However, $\mathbb{E}[Y | X, S, A = j]$ by definition refers to the (conditional) expected reward rather than the outcome of regression. $\mu_j(X, S) \approx \mathbb{E}[Y | X, S, A = j]$ should be more appropriate (i.e., using "$\approx$" instead of "$=$").

- Reading the Introduction, it was unclear whether the paper was focusing on user- or action-side fairness. Since there are some discussions on item fairness in recommender systems (e.g., Singh & Joachims, 18), it might be helpful to explicitly mention either sensitive features belonging to users' covariate or actions' features. (In Section 2, I soon realized that the paper focuses on user-side fairness, though)

Singh & Joachims, 18: Ashudeep Singh, Thorsten Joachims. "Fairness of Exposure in Rankings". KDD, 2018.

---

> ### Author Response · Authors · 2023-11-21
> **Response to Reviewer 3M28 #1**
>
> Thank you for your comprehensive and actionable review! We took all your comments at heart and improved our paper as follows:
>
> ## Response to “Weaknesses”
>
> 1. Formulations and mathematical descriptions: Thank you for pointing this out. Please refer to our “Response to Questions”.
> 2. Related work on fair representation learning: We kindly refer to Appendix A.1, in which we provide a brief overview of related fair representation learning. We would like to emphasize that our main contribution is **not** a new method for fair representation learning. Rather, we adapt fairness notions and provide an **understanding of how these fairness notions interact in the context of off-policy learning**, while we at the same time propose a **practical framework to learn fair optimal policies and provide theoretical guarantees** in the form of generalization bounds. However, we acknowledge that the literature stream on fair representation learning deserves a more thorough treatment in our related work.
>
>     **Action:** We **expanded our related work on fair representation learning and moved it to the main paper** (see Sec. 2, new paragraph “fair representation learning”). Furthermore, we **added new experiments** using alternative approaches for fair representation learning (see our **new Appendix K**). However, we emphasize that fair representation learning is orthogonal to our contribution as we develop a new framework on top of it for fair off-policy learning.
>
> 3. How is the problem of off-policy learning resolved? Thank you for giving us the opportunity to clarify this point. You are correct that one of the key challenges in combining off-policy learning with our fairness constraints is the problem of estimating the conditional policy values $V^\mathrm{m}_s(\pi)$. In our paper, **we proposed unbiased estimators** $\hat{V}^\mathrm{m}_s(\pi)$ (see Sec. 5.2.), which can be used in combination with any standard off-policy learning method $\mathrm{m} \in \{\mathrm{DM}, \mathrm{IPW}, \mathrm{DR}\}$. In our experiments, we **provided ablation studies** in the sense that we applied our framework with all three methods $m$ (see e.g., Table 1). Furthermore, we **provided theoretical guarantees** for the finite-sample errors of our fairness objective estimators in the form of generalization bounds (Theorem 1), which depend on the method $m$ (through the constant $K_m$). Hence, **we provided both theoretical and empirical insights for solving the off-policy learning problem** in our fairness context.
>
>     **Action:** We clarify the above in both Sec. 5.2. and Sec. 5.3. Furthermore, we emphasize the dependence of the bounds in Theorem 1 on the off-policy learning method $m$.
>
> 4. Results on real-world data: In Table 2, we first report the results for our framework with only action fairness and without value fairness (second row, “FairPol only with action fairness”). Here, value fairness is indeed worse compared to the unrestricted policy (first row). However, this is to be expected as FairPol enforces action fairness, which can worsen value fairness (we refer to our toy example in Appendix C. for a detailed discussion on this issue). In contrast, the (estimated) action fairness is improved by a large margin. Furthermore, comparing our results for value fairness (third and fourth row of Table 2) directly with the unrestricted policy (first row) is misleading because the unrestricted policy does not ensure action fairness. Instead, FairPol with max-min fairness (fourth row of Table 2) is effective in improving value fairness compared to FairPol with only action fairness (second row of Table 2). In summary, **the results using real-world data confirm the effectiveness of our framework.**
>
>     **Action:** We followed your suggestion and expanded our discussion of our results using real-world data in Sec. 6.
>
> 2. Clarity of writing: Thank you for pointing this out. We improved our writing along your suggestions. Please refer to our “Response to Questions”.

---

> ### Author Response · Authors · 2023-11-21
> **Response to Reviewer 3M28 #2**
>
> ## Response to “Questions"
>
> 1. Definition of action fairness: Thank you for critically questioning our definition of action fairness and for allowing us to elaborate on our definition. You are correct that in our definition of action fairness ($\pi(X) \perp S$), our policy $\pi$ is a function of all confounders $X$, i.e., also the ones that may be spuriously correlated with $S$. However, enforcing the independence statement $\pi^{\mathrm{af}}(X) \perp S$ ensures that the policy $pi$ does not depend on such spuriously correlated confounders. To see this, let us consider the example you suggested, where $X_0 = f(S)$ for some (measurable) function $S$ and $X_{1:m}$ are independent of $S$. This implies that $\pi(X) = \pi(f(S), X_1, \dots, X_m)$ is a (measurable) function of $S$. Hence, the independence $\pi(f(S), X_1, \dots, X_m) \perp S$ only holds if either $f$ is constant (in which case $X_0$ and $S$ are independent), or if $\pi(\cdot, x_1, \dots, x_m)$ is constant for all x_1, \dots, x_m (almost surely). We agree that the definition you provided would be another valid way of defining action fairness. However, the benefit of our definition is that we do not assume prior knowledge on $X$ (e.g., which components are independent of $S$).
>
>     **Action:** We added a clarification to Sec 4.1 along the above lines.
>
> 2. Pareto optimal policies: In our paper, we go beyond existing work from Viviano and Brandic (VB) in several ways: (i) We consider different fairness notions than VB (i.e., action fairness, value fairness, and interactions thereof); (ii) VB maximize fairness over the set of Pareto optimal policies. Here, Pareto optimality is defined so that the policy value of one sensitive group cannot be improved without reducing the policy value for the opposite group. In contrast, we propose to incorporate our fairness notions by adjusting the off-policy learning objective (value fairness), and then maximize this objective over the class of action fair policies. (iii) While VB is restricted to learning linear policies, our approach (outlined in (ii)) enables learning arbitrarily _non-linear_ policies (which we refer to as “general policies”). This is possible because we can incorporate the action fairness constraint by leveraging fair representation learning to obtain a representation independent of the sensitive attribute, which can be used in a second step to train an optimal policy. Notwithstanding, the VB approach can **not** work with non-linear policies and uses different fairness notions, hence, is **not** applicable in our case.
>
>     **Action:** We clarified the definition of “Pareto-optimality” and “general policy” in our related work and elaborated on the differences in our paper.
>
> 3. Experimental details: **In Table 1, we reported the results for all methods** $\mathrm{m} \in \{\mathrm{DM}, \mathrm{IPW}, \mathrm{DR}\}$. Here, for each method, we reported three rows corresponding to $m$. In particular, we do not observe large differences in results between the three methods. Thus, we fixed $m = \mathrm{DR}$ (corresponding to the doubly robust method) in our other experiments (e.g., Table 2; Fig. 3. and 4.).
>
>     **Action:** We added a clarification to Sec. 6.
>
>
>
> ## Response to “Other suggestions”
>
>
>
> * “Outcome regression”: Thank you for pointing out this ambiguity. **Action:** We replaced the naming “outcome regression” with “response surface”, which is consistent with previous work on off-policy learning and treatment effect estimation.
> * User vs action covariates: Thank you. You are correct in that we defined our covariates $X$ to be user-specific attributes, not attributes of the action/treatment. **Action:** We added a clarification to Sec. 3.

---

> ### Comment · Reviewer_3M28 · 2023-11-22
>
> Thank you for the detailed response and for sharing the revision. The revision on the discussion of fair representation learning and comparison with (Viviano and Brandic) would be helpful for readers.
>
> I have several remaining comments on rebuttals and revisions.
>
> ---
>
> - According to the call for papers, we have a strict page limit of 9 pages. However, the current draft is over this page limit. How do you plan to fit the paper to this limit?
>
> > There will be a strict upper limit of 9 pages for the main text of the submission, with unlimited additional pages for citations. This page limit applies to both the initial and final camera ready version. https://iclr.cc/Conferences/2023/CallForPapers
>
> ---
>
> - About Weakness and Question 1
>
> > 1. Definition of action fairness: Thank you for critically questioning our definition of action fairness and for allowing us to elaborate on our definition. You are correct that in our definition of action fairness ($\pi(X | S)$), our policy $\pi$
>  is a function of all confounders $X$, i.e., also the ones that may be spuriously correlated with $S$.
>
> In this case, the following sentence in the current draft should be mathematically incorrect, as I pointed out in the initial review. This part should be revised to avoid readers' confusion about math in this paper.
>
> > a policy $\pi$ should assign the same action $A = A'$ to two individuals with the same covariates $X=X'$ but different gender $S \neq S'$.
>
> I am not pointing out the description of Definition 1. Rather, the concept of describing action fairness (the above sentence) is contradictory to the description in Introduction.
>
> > We agree that the definition you provided would be another valid way of defining action fairness. However, the benefit of our definition is that we do not assume prior knowledge on $X_0$ (e.g., which components are independent of $S$).
>
> In addition, I do not suggest assuming prior knowledge on $X_0$ here. My point is that the proper description, that aligns with Introduction or some other sentences in the paper,  should need $X_0$, regardless of assuming prior knowledge on $X_0$.
>
> ---
>
> - About suggestion and revision
>
> > “Outcome regression”: Thank you for pointing out this ambiguity. Action: We replaced the naming “outcome regression” with “response surface”, which is consistent with previous work on off-policy learning and treatment effect estimation.
>
> I was not commenting on the terminology but was pointing out the improper use of math. By definition, $\mathbb{E}[Y | X, S, A=j]$ is by the expected outcome conditioned on $X, S, A=j$. Therefore, if the paper denotes $\mu_j(X, S) = \mathbb{E}[Y | X, S, A=j]$, $\mu_j(X, S)$ cannot be the regressed outcome (because it is the "true" expected outcome by definition) in DM and DR. Though I haven't checked all the math descriptions, the math in this paper seems doubtful to me, given that there are already two misrepresentations.
>
> ---
>
> - About theoretical analysis.
>
> This is a clarification question, where are $\nu$ and $\eta$ defined in Theorem 1? (It seems they are not defined.)
>
> > Furthermore, we provided theoretical guarantees for the finite-sample errors of our fairness objective estimators in the form of generalization bounds (Theorem 1), which depend on the method $m$ (through the constant $K_m$).
>
> In addition, several things remain unclear, e.g., (i) how is the source of difference in the generalization bound among DM, IPS, and DR? In other words, in what situation does the difference emerge? (ii) How the generalization bound can be different when considering fairness compared to standard analysis of the generalization bound of DM, IPS, and DR? (iii) How to interpret three results of generalization bounds? It would be great if the presentation of the theoretical analysis could be improved so readers can understand the connections and differences with the standard OPL papers.
>
> ---
>
> - About the experiment.
>
> >  we provided both theoretical and empirical insights for solving the off-policy learning problem in our fairness context.
>
> Typical OPL papers (e.g., Swaminathan & Joachims, 15) often compare the methods with varying data sizes to confirm the theoretical results of generalization bound analysis, while not provided in this paper.
>
> Overall (not limited to the experiment section), the discussion of challenges in using observational data still seems thin from the manuscript despite being one important topic of this paper.
>
> Swaminathan & Joachims, 15: Adith Swaminathan and Thorsten Joachims. "The Self-Normalized Estimator for Counterfactual
> Learning." NeurIPS, 2015.

---

> ### Author Response · Authors · 2023-11-22
>
> Thank you for your quick response and for allowing a constructive discussion. In the following, we address your remaining concerns and are confident that this improved our paper further.
>
> * **Page limit:** Thank you for pointing this out. We were unsure whether the page limit applies during the rebuttal, and, to address your comment, we uploaded a revised version with exactly 9 pages.
> * **Action fairness:** We apologize for the misunderstanding. You are correct, our sentence in the introduction does not fit our formalization of action fairness. **Action:** We changed our introduction to action fairness at the beginning of Sec. 4. to fit our formal definition.
> * **Outcome regressions/response surfaces:** You are correct that response surfaces $\mu_j(X, S)$ denote the ground-truth expectation of $Y$ conditioned on $X, S, A = j$. This is deliberate because the response surfaces $\mu_j(X, S)$ do not denote estimated quantities. However, this conditional expectation can be estimated from the observational data (e.g., by performing a regression of Y on X, S, A.). Note that the same holds for the ground-truth behavioral policy $\pi_b$, which reduces to a classification problem for binary treatments. In our experiments, we use a TARNet [1] to estimate the nuisance parameters (i.e., $\mu_j$ and $\pi_b$). However, our framework is agnostic to the nuisance estimator and could be used in combination with any other estimator (e.g., S/T learner [2]).
>
>     Hence, we emphasize that **all our mathematical statements are correct.** Please note that off-policy learning/ treatment effect estimation under nuisance parameters is a well-studied topic and that we are not the first who propose this type of notation [2, 3,4]. Is your concern potentially that, in Theorem 1, we do not account for estimation errors in the nuisance parameters (because we use the ground-truth nuisance parameters)? While this is true, it is not a major concern because it has been shown that (under uniform consistency of nuisance estimators), estimation errors in the nuisance parameters do not affect the leading term in the convergence rate (see Sec. 3.1, Sec.3.2, Lemma 4 in [4]). This is also consistent with previous work that provides generalization bounds for off-policy learning [5].
>
> .

---

> > ### Author Response · Authors · 2023-11-22
> >
> > * **Theoretical analysis:** We apologize for the missing definitions $\nu$ and $\eta$. These were defined in our boundedness Assumption 2, which we moved to the appendix to space constraints. Unfortunately, we did not realize that this would leave $\nu$ and $\eta$ undefined in the main paper.  Of note, similar boundedness assumptions are standard in the off-policy learning/treatment effect literature [2, 3, 4].  \
> > **Action:** We moved Assumption 2 to Sec. 5.3. of the main paper.
> >
> > Further clarifications:
> > * “In what situation does the difference between the policy learning methods emerge?” This is an interesting question. One example where the differences emerge is under limited overlap, i.e., when certain individuals are almost always/never treated. In this case, the propensity score $\pi_b$ is close to 0 or 1. Hence, $\eta$ becomes very small such that $K_{IPW}$ and $K_{DR}$  become large but $K_{DM}$ stays constant. Intuitively, this is because both IPW and DR use division by the propensity score $\pi_b$ and $(1 - \pi_b)$ (see Eq. 3 and 4), which becomes unstable once $\pi_b$ is close to 0 or 1.
> > * Comparison with standard bounds: Here, we kindly refer to Eq.(9), which is precisely the bound for the optimal unrestricted policy (i.e., standard off-policy learning without fairness considerations).
> > * Interpretation and implications: Theorem 1 has two main qualitative implications: (i) **We can achieve a $\sqrt{n}$-convergence rate for all fairness objectives** whenever we optimize over a model class $\Pi$ with $\sqrt{n}$-vanishing Rademacher complexity, such as neural networks. (ii) Compared to the bound for the unrestricted policy (Eq. 9), the bounds corresponding to our fairness objectives depend on $\nu$ and hence on the population balance within the marginal distribution of the sensitive attribute $S$. This implies that the **bounds become loose whenever a sensitive group is underrepresented in the data.** For example, if the data mostly contains male individuals and only a few female individuals, the statistical estimation error of our fairness objective is expected to increase whenever gender is the sensitive attribute. Hence, **our theory has practical implications for guiding decision-makers who consider applying our framework in real-world situations.**
> >     * **Action:** We expanded our discussion on Theorem 1 in Sec. 5.3.
> >
> > * **Experiment:**
> >     * Varying sample sizes: Thank you for the suggestion to add experiments with varying sample sizes.  \
> > **Action:** We repeat our experiments from Table 1 with different sample sizes and report the results **in our new Appendix M.**
> >     * Observational data: We would like to emphasize that observational data is central to our paper. Here, we follow well-established literature on off-policy learning from observational data built upon the most common policy learning methods (DM, IPW, DR), and provide theoretical results. Furthermore, we empirically test the sensitivity of our experimental results w.r.t. Estimation errors within the nuisance parameters (see our **new Appendix L**).
> >
> >
> > ## References
> >
> > [1] Shalit et al. (2017). “Estimating individual treatment effect: generalization bounds and algorithms” ICML.
> >
> > [2] Curth et al. (2021). “Nonparametric Estimation of Heterogeneous Treatment Effects: From Theory to Learning Algorithms” AISTATS
> >
> > [3] Kallus (2021) “More Efficient Policy Learning via Optimal Retargeting”. Journal of the American Statistical Association.
> >
> > [4] Athey and Wager (2021). “Policy learning from observational data” Econometrica.
> >
> > [5] Hatt et al. (2022). “Generalizing off-policy learning under sample-selection bias” UAI

---

> > > ### Comment · Reviewer_3M28 · 2023-11-23
> > >
> > > Thank you for the response. The comparison of the theoretical analysis with standard OPL seems understandable and would be valuable for the readers.
> > >
> > > I will raise my score to 6. However, I should also note that my evaluation of this paper is exactly on the borderline, and this is a very weak signal.
> > >
> > > I encourage the authors to improve the presentation of the paper further, particularly about the OPL part. I would also recommend double-checking mathematics and notations more than once before publication.

---

> > > > ### Author Response · Authors · 2023-11-23
> > > > **Thank you for raising your score**
> > > >
> > > > Thank you for your response and for raising your score! We are happy that we could clarify the remaining open points.
> > > >
> > > > To improve our presentation, we added another remark to our OPL section, where we explicitly say that we use ground-truth nuisance parameters in our notation. As you suggested, we will carefully proofread our paper and the mathematical details again before potential publication. We remain open to further suggestions or queries you might have. Please let us know if any additional points come to your attention.

---

### Official Review · Reviewer_TRGr · 2023-10-30

**Soundness:** 3 good
**Presentation:** 3 good
**Contribution:** 2 fair
**Rating:** 5
**Confidence:** 3

**Summary:**

The paper studies off-policy learning under two fairness notions: (1) action fairness that requires the actions made by policies to be independent of sensitive attributes; and (2) max-min fairness and envy-free fairness defined based on utility (policy value) perceived by groups. The paper first explores the relations between action fairness and policy value fairness, and shows that the two notions are compatible under certain conditions. The paper then develops an algorithm that learns fair optimal policies. Experiments on both synthetic and real data validate the proposed algorithm.

**Strengths:**

1. The paper simultaneously consider action fairness and policy value fairness in off-policy learning. The results of relations between two notions of fairness are novel to my knowledge.
2. The paper is well-organized and easy to follow. In addition to the algorithm, it provides generalization bound and validation on both synthetic and real data.

**Weaknesses:**

1. While the proposed method can learn optimal policies under both action and policy value fairness, the underlying technique seems to be very similar to the existing methods. The technical contribution of this paper is unclear to me. Specifically, the proposed solution includes two steps: the first step aims to learn fair representation to ensure action fairness, while the second step incorporates the policy value fairness constraints to the objective function. For the first step, the idea of leveraging adversarial objectives to achieve independence between sensitive attribute and representation is similar to the concept of GAN, a common approach for fair representation learning. For the second step, the learning objective with fairness constraint is also straightforward.

2. The paper doesn’t compare with any algorithms in experiments. While the authors claim that existing methods are proposed in a different setting and cannot serve as the baseline, I wonder whether the methods can be adjusted and applied. For example, when learning fair representation in the first step, many approaches in fair representation learning may be applied and serve as baselines.

**Questions:**

1. Many approaches have been proposed to learn ML models under fairness constraints such as demographic parity, envy-free fairness, max-min fairness. Except for the objective function, what are the technical challenges of off-policy learning compared to tractional supervised learning? In other words, why cannot methods proposed in fair supervised learning be modified and directly adapted to off-policy learning?

2. Can we adapt existing methods to your setting and treat them as baselines?

---

> ### Author Response · Authors · 2023-11-21
> **Response to Reviewer TRGr #1**
>
> # Response to reviewer TRGr
>
> Thank you for your review and your helpful comments!
>
>
> ## Response to “Weaknesses”
>
> * Thank you for giving us the opportunity to clarify our technical contributions. You are correct in that we build our framework on existing literature streams, e.g., fair representation learning. However, we believe that our paper makes several non-trivial technical contributions beyond that. In particular: (i) We adapt fairness notions from related literature and **formalize these for off-policy learning from observational data**. (ii) We provide a **theoretical understanding of how these fairness notions interact in the context of off-policy learning** (see Lemma 1 and 2, as well as our toy examples in Appendix C and D). (iii) We propose a **practical framework to learn fair optimal policies. **While we build upon ideas from fair representation learning to achieve action fairness, we combine these with the off-policy learning methodology** and provide novel theoretical guarantees** in the form of generalization bounds (Theorem 1). Hence, we believe that our paper contains several technical contributions that add to the state-of-the-art in algorithmic fairness and off-policy learning in a non-trivial manner.
>
>     **Action:** We clarified our technical contributions in Sec.1. We also expanded our discussion (see end of Sec. 6), in which we also reiterated our contributions
>
> * **No baselines available:** We agree that, generally, a thorough comparison with baselines is beneficial to provide a rigorous empirical evaluation. Unfortunately, there are **no** baselines directly available as our setting of fair policy learning has not yet been studied before.
> Nevertheless, we took great strides to adapt existing methods as baselines. In the context of our work, we see two potential literature streams that could provide methods baselines for our experiments: (i) Methods for standard off-policy learning from observational data (not considering fairness); and (ii) methods that incorporate fairness notions into off-policy learning. In the following, we lay out how we build upon methods from (i) and (ii) for benchmarking:
>     * For (i), the following three methods are considered standard in the literature [1]: the direct method, the IPW method, and the DR method. **Our proposed framework works for all of these, and we provided experiments for all three baselines **(see Table 1). We did not observe large differences in performance.
>     * For (ii), **we are not aware of any other work that aims at achieving the same fairness notions (i.e., combining action and value fairness) in the context of off-policy learning**. Any potential benchmarking experiment with existing methods would thus involve comparing methods that satisfy different fairness notions. Hence, **we follow established fairness literature [e.g., 2, 3] and do not compare methods targeted at different fairness notions**. Instead, we investigated in our experiments how well our proposed method achieves our proposed fairness notions, and how this affects the performance (policy value). For this purpose, **we provided several ablations studies:** for example, we remove parts of our method (value fairness), and **we compare with an unrestricted policy** that does not satisfy any fairness notions.
>
>     We agree that our approach for fair representation learning (adversarial domain confusion loss) could in principle be replaced by any other approach that aims at enforcing independence from the sensitive attribute.  Hence, we provide an **additional ablation study (see new Appendix K)**, where we compare the first step of our framework with two other approaches: (i) Adversarial learning using gradient reversal; and (ii) regularization using a Wasserstein distance. Our chosen approach (adversarial domain confusion loss) performs best, which is also consistent with prior literature [1].
>
>
>     **Action:** We **added new experiments** based on fair representation learning (see our **new Appendix K**).

---

> > ### Author Response · Authors · 2023-11-21
> > **Response to Reviewer TRGr #2**
> >
> > ## Response to “Questions”
> >
> > * **Differences to standard supervised learning:** Thank you for the question. As you mentioned correctly, one important difference between off-policy learning and (standard) supervised learning is the **difference in objective function**: In off-policy learning, we are interested in maximizing the policy value, which is estimated using nuisance parameters that depend on the sensitive attribute. Note that we can not simply remove dependence from the sensitive attribute, as this could lead to unobserved confounding and thus biased policy value estimates. Hence, combining off-policy learning with fairness results in challenges for both (i) fairness considerations; and (ii) finite-sample performance when introducing fairness constraints during policy learning.
> >
> >     In our paper, we address these challenges as follows: For (i), we provide a **theoretical understanding** of different fairness notions in off-policy learning (Lemma 1 and 2). We also show explicitly how “action-fair” policies may still be considered unfair due to discrepancies in policy values for different sensitive groups (see our toy examples in Appendix C and D). For (ii), we investigate the off-policy learning problem with fairness constraints from a statistical perspective and provide theoretical learning guarantees in the form of generalization bounds (Theorem 1).
> >
> > * **Additional baselines:** Thank you for your question. We agree that, generally, a thorough comparison with baselines is beneficial to provide a rigorous empirical evaluation. Unfortunately, there are **no** baselines directly available. Nevertheless, we took great strides to adapt existing methods as baselines where possible (see our “Response to Weaknesses”, point 2). We also **added new experiments** using benchmarks from fair representation learning (see our **new Appendix K**).
> >
> >
> > ## References
> >
> > [1] Melnychuk et al. (2022). “Causal transformer for estimating counterfactual outcomes” ICML.
> >
> > [2] Madras et al. (2018). “Learning adversarially fair and transferable representations” ICML
> >
> > [3] Craeger et al. (2019) “Flexibly Fair Representation Learning by Disentanglement” ICML

---

### Official Review · Reviewer_1vxi · 2023-11-01

**Soundness:** 3 good
**Presentation:** 4 excellent
**Contribution:** 3 good
**Rating:** 8
**Confidence:** 3

**Summary:**

This paper introduces a framework for fair off-policy learning that is a neural approach to fair off-policy learning (in comparison to related works that are only linear). The framework is broken up into two stages. The first deals with action fairness (the action should be independent of a sensitive attribute). The second adds policy value fairness for different sensitive groups (the authors used two variants: envy-free fairness and max-min fairness). The authors provide generalization bounds and perform experiments on synthetic data and real world medical insurance data.

**Strengths:**

The paper is very well written and easy to follow. The authors take considerable care when defining concepts in the paper to make things clear for the reader.

The paper is novel in that it produces a framework that is considerably less restrictive than other related works. Specifically, the addition of a neural approach to fair off-policy learning. It is also general enough to fit many different contexts and needs of practitioners.

The authors provide clear generalization bounds.

The authors provide clear experimental results and describe the insights well.

**Weaknesses:**

Mostly minor nits for weaknesses:

It is unfortunate that no other baselines are available for this work.

Although space is very limited, it would be good to include more discussion from Appendix I in the paper.

The plots in the paper need to be more readable. Thicker lines and larger text to match the text size in the paper.

**Questions:**

See above weaknesses

---

> ### Author Response · Authors · 2023-11-21
> **Response to Reviewer 1vxi**
>
> Thank you for your positive evaluation of our paper! We took all your comments at heart and improved our paper accordingly.
>
> ## Response to “Weaknesses”
>
> * **No baselines available:** We agree that, generally, a thorough comparison with baselines is beneficial to provide a rigorous empirical evaluation. Unfortunately, there are **no** baselines directly available as our setting of fair policy learning has not yet been studied before.
> Nevertheless, we took great strides to adapt existing methods as baselines. In the context of our work, we see two potential literature streams that could provide methods baselines for our experiments: (i) Methods for standard off-policy learning from observational data (not considering fairness); and (ii) methods that incorporate fairness notions into off-policy learning. In the following, we lay out how we build upon methods from (i) and (ii) for benchmarking:
>     * For (i), the following three methods are considered standard in the literature [1]: the direct method, the IPW method, and the DR method. **Our proposed framework works for all of these, and we provided experiments for all three baselines** (see Table 1). We did not observe large differences in performance.
>     * For (ii), **we are not aware of any other work that aims at achieving the same fairness notions (i.e., combining action and value fairness) in the context of off-policy learning**. Any potential benchmarking experiment with existing methods would thus involve comparing methods that satisfy different fairness notions (and thus have entirely **different** objectives that are **incompatible** and thus **not** comparable).
> Hence, **we follow established fairness literature [e.g., 2, 3] and do not compare methods targeted at different fairness notions**. Instead, we investigated in our experiments how well our proposed method achieves our proposed fairness notions, and how this affects the performance (policy value). Hence, **we provided several ablations studies:** for example, we removed parts of our method (value fairness), and **we compared with an unrestricted policy** that does not satisfy any fairness notions.
>
>     Nevertheless, we understand that you ask for a more extensive numerical evaluation.
>
>     **Action:** Our approach for fair representation learning (adversarial domain confusion loss) can in principle be replaced by any other approach that aims at enforcing independence from the sensitive attribute.  Hence, we provide a **new ablation study (see new Appendix K)**, where we compare the first step of our framework with two other approaches: (i) Adversarial learning using gradient reversal; and (ii) regularization using a Wasserstein distance. Our chosen approach (adversarial domain confusion loss) performs best, which is also consistent with prior literature [4].
>
> * **Limited discussion:** Thank you. We followed your suggestion and, to improve our paper, now offer **an extended discussion**.
>
>     **Action:** We followed your suggestion and moved parts of Appendix I to our **new Sec. 7**. In particular, we now discuss our contribution and provide an outlook on future research directions.
>
> * **Readability of plots:** Thank you; we have improved our plots. **Action:** We followed your suggestion and increased both line thickness and font size in Fig.3 and 4.
>
>
> ## References
>
> [1] Athey and Wager (2021). “Policy learning from observational data” Econometrica.
>
> [2] Madras et al. (2018). “Learning adversarially fair and transferable representations” ICML
>
> [3] Craeger et al. (2019) “Flexibly Fair Representation Learning by Disentanglement” ICML
>
> [4] Melnychuk et al. (2022) "Causal transformer for estimating counterfactual outcomes" ICML

---

### Official Review · Reviewer_mTnw · 2023-11-09

**Soundness:** 2 fair
**Presentation:** 2 fair
**Contribution:** 2 fair
**Rating:** 5
**Confidence:** 4

**Summary:**

This paper studies offline policy learning from observational data with some fairness objectives. Specifically, they proposed two fairness objectives. The first one is called "action fairness" which aims to ensure that the learned policy should be independent of the sensitive attributes. The second one is called "value fairness" and there are two variants: "envy-free fairness" which aims to make the policy value the same across sensitive attributes and "max-min" fairness which aims to maximize the minimum policy value across sensitive attributes. They proposed a machine learning algorithm to achieve these fairness objectives in two steps. The first step is to achieve "action fairness". The paper borrows ideas from prior work to learn representations that are not predictive of sensitive attributes while maintaining predictive power of outcome using some loss functions including the confusion loss. The second step is to learn a policy using empirical counterfactual estimate of the policy value for each sensitive attribute with respect to the two proposed "value-fairness" objectives. They quantify the generalization error of the proposed algorithm, and conducted empirical evaluation on both simulated and real-world data.

**Strengths:**

1. Fairness in offline policy learning from observational data seems an interesting and important problem.

2. The paper is, for the most part, well-written and easy to follow.

3. They conducted both theoretical analysis and empirical evaluation on the proposed algorithm.

**Weaknesses:**

1. One concern is that the paper does not discuss how they deal with observational data. In particular, the proposed method depends on several estimators like DM, IPW, DR. But some quantities are unknown in the observational setting, like \mu_1, \mu_0, \pi_b. I would be great if the authors can discuss how these quantities are obtained, and how they affect the theoretical and empirical results.

2. For empirical evaluation, especially for real-world data one, we do not know the counterfactual outcome of counterfactual actions. It would be great if the authors can discuss how the performance metrics are calculated.

3. The contribution seems limited. The fairness notions proposed in this paper appear in many prior works. The approach to achieving "action fairness" seems to be borrowed from prior works. The approach to achieving "value" fairness is a direct application of counterfactual estimator.

4. It would be great if the authors could discuss the advantages and disadvantages of these fairness objectives. In particular, when are these fairness objectives applicable? For example, envy-free does not seem to make sense to me in many applications, since it might happen that it is very easy to achieve larger expected outcome for one group than the other. Maybe the maximum value we can achieve for group A is 1, and the minimum value we can achieve for group B is 1 and the maximum for B is 2. Suppose there is a policy that can achieve value 1 for A and 2 for B, why would we decrease the value for B just to satisfy this fairness notation? For max-min fairness, similar arguments apply. Since group A is always lower in policy value, we will only learn our policy for group A. Why is this advantageous compared to fairness definitions that achieve pareto optimality?

Some minor points:
4. In assumption 1, it should be P(A=1 | X=x, S=s), right? Otherwise, equation (3) and (4) might be ill-defined since \pi_b(X,S) can be zero.

5. Equation (2), (3), (4), \phi is a function of D, which contains i.i.d. random variables of several datapoints. But in the right hand side, there is only one datapoint. It is a bit confusing to me.

6. Theorem 1 is hard to understand. I think it must depend on the logging/behavior policy. Maybe it is captured in K? It would be great if the authors can have a clear definition of these notations. And it does not seem to take into account the fact that we are in the observational setting as I mentioned in 1.

**Questions:**

See weakness.

**Details Of Ethics Concerns:**

See weakness 4.

---

> ### Author Response · Authors · 2023-11-21
> **Response to Reviewer mTnw #1**
>
> Thank you for your review and your helpful comments. As you can see below, we have carefully revised our paper along with your suggestions.
>
>
> ## Response to “Weaknesses”
>
> 1. **Observational data/ nuisance parameters:** Thank you for your question. You are correct that our method (or rather any method for off-policy learning) depends on the policy score $\psi^\mathrm{m}(\pi, \mathcal{D})$ and, therefore, on the unknown nuisance parameters $\mu_1$, $\mu_0$, and $\mu_0$ (which correspond to the choice of $\mathrm{m} \in \{\mathrm{DM}, \mathrm{IPW}, \mathrm{DR}\}$, respectively). Under Assumption 1, these nuisance parameters can be estimated from the observational data using arbitrary regression/machine learning algorithms (see Sec. 3). **In our experiments on synthetic data, we do not focus on the question of estimation errors in the nuisance parameters but rather aim to explore the impact of our proposed fairness notions**. For this purpose, we refrain from estimating the nuisance parameters but instead use the ground-truth nuisance parameters, which are known in our simulations (we refer to Appendix G for details). Note that we do this for all methods to ensure a fair comparison of the results.
>
>     Motivated by your comment, we added **new experimental results** (see our **new Appendix L**). Therein, we re-ran our experiment from Table 1 but now use TARNet as a state-of-the-art method [1] to estimate nuisance parameters (instead of using the ground-truth). This allows us to analyze the robustness of our results on synthetic data with respect to nuisance errors. Our new results show that our results **remain robust** when using estimated nuisance parameters.
>
>     **Action:** We provided a more elaborate explanation for our evaluation. Furthermore, we **added new experimental results** by using TARNet for estimating nuisance parameters instead of using the ground truth (see our **new Appendix L)**. We added details regarding the implementation of TARNet to Appendix M.
>
> 2. **Empirical evaluation on real-world data:** Thank you for your question. You are correct that we do not observe the counterfactual outcomes on the real-world data, which means that we can **not** evaluate our methods using oracle metrics (as in Table 1).  As mentioned in the previous point, we use a TARNet to estimate the nuisance parameters on real-world data. Thus, we can plug the estimated nuisance parameters into our (conditional) policy value estimators $\hat{V}^\mathrm{m}(\pi)$ and  $\hat{V}^\mathrm{m}_s(\pi)$ (proposed in Sec. 5.2.). This is what we report in, e.g., Table 2. Of note, our evaluation is in line with works on treatment effect estimation, which often report the factual MSE as a heuristic to measure performance on real-world data [1]. In principle, our policy value estimators may be biased if not all confounders are collected in the data. However, in our experiments, we do not aim to compare performance (e.g., in terms of unbiased estimation), but rather aim to investigate the impact of our fairness notions on our learned policy.
>
>     For action fairness, we reported the Spearman correlation between policy predictions and the sensitive attribute, which enables detecting (potentially nonlinear) dependence.
>
>
>     **Action:** We added a clarification to Sec. 6.
>
> 3. **Contribution/novelty of fairness notions:** Thanks. You are correct in that our proposed fairness notions are adapted from existing literature streams. For example, “action fairness” resembles “demographic parity” in standard fair machine learning, and both “max-min” and “envy-free” fairness have also been proposed in works from optimization (e.g., from resource allocation). The reason is that the fairness notions are deeply grounded in distributive justice. Please note that **we do not claim to be the first to propose the mentioned fairness notions**. Instead, our contributions are as follows:
>     1. (i) We propose to adapt existing fairness notions from related literature (e.g., resource allocation), but **formalize these for off-policy learning from observational data**;
>     2. (ii) We provide a **theoretical understanding of how these fairness notions interact in the context of off-policy learning** (see Lemma 1 and 2, as well as our toy examples in Appendix C and D);
>     3. (iii) We propose a **practical framework to learn fair optimal policies and provide theoretical guarantees** in the form of generalization bounds.
>
>     **Action:** We clarified our contributions in Sec.1. We also expanded our discussion (see end of Sec. 6), in which we also reiterated our contributions

---

> > ### Author Response · Authors · 2023-11-21
> > **Response to Reviewer mTnw #2**
> >
> > 4. **Advantages and disadvantages of fairness notions:** Thank you for giving us the opportunity to clarify the applicability of our fairness notions. First of all, we would like to emphasize that we propose a **joint approach that learns both action-fair and value-fair policies**. For example, max-min fairness only provides a reasonable fairness notion in combination with action fairness (we refer to Lemma 1 for details).
> >
> >     **A key benefit of action-fair policies is that the corresponding policy predictions do not depend on the sensitive attribute**. This is of great importance in various domains where legal restrictions prohibit direct discrimination, e.g., credit lending [5]. However, action-fair policies may still be unfair in the sense that they result in discrepancies in policy values between the sensitive groups (value unfairness). We refer to Appendix C for a detailed example, in which we show that action fairness can even increase value unfairness. As a remedy, we propose two value fairness notions (see below). There are several key benefits of value-fairness notions that offer several advantages: (i) envy-free fairness is useful whenever the discrepancies in policy values should be directly controlled (e.g., due to legal regulations); (ii) max-min fairness is useful in high-stakes situations whenever the worst-case policy value should be optimized; (iii) both fairness notions can be easily combined with action fairness by using our FairPol framework. We agree that both envy-free fairness and max-min fairness have limited applicability in the provided toy example (e.g., when the policy values of the sensitive groups are on different scales). However, **we do not claim that these fairness criteria should always be applied but recommend careful consideration using the specific characteristics and constraints of any practical application.** This is in line with much of the literature on algorithmic fairness, recommending that fairness notions must be chosen carefully to fit the specific context [5].
> >
> >     **Action:** We **added a new, detailed discussion** with practical considerations (see our **new Appendix J**). Therein, we provide an overview of the applicability of our three proposed fairness criteria (action fairness, envy-free fairness, and max-min fairness). We further discuss the advantages and disadvantages of the different fairness criteria, and make practical recommendations under which settings the different fairness criteria are suitable in practice. In particular, we list examples of real-world situations and legal frameworks that make specific suggestions as to which fairness criteria practitioners should choose.
> >
> > ## Response to “Minor points”
> >
> > 1. Thank you. You are correct: it should be $\mathbb{P}(A=1 \mid X = x, S = s)$. We apologize for the typo and corrected it in our revised paper.
> > 2. In our paper, we defined $\mathcal{D} = (X, S, A, Y)$, where $(X, S, A, Y) \sim \mathbb{P}$ is distributed according to the observational population. Hence, **$\mathcal{D}$ does **not** contain i.i.d. datapoints**. **Action:** We admit that this notation may be confusing as $\mathcal{D}$ is often used to denote observational datasets. Hence, for clarification, we replaced  $\mathcal{D}$ with $W$ in our revised paper.
> > 3. Thank you. You are correct: our bounds in Theorem 1 do indeed depend on the nuisance parameters (in particular on the logging policy/propensity score) through the constant $K_m$, where $\mathrm{m} \in \{\mathrm{DM}, \mathrm{IPW}, \mathrm{DR}\}$ indicates the policy score (the thus the choice of nuisance parameters). In our proof (see Appendix B), we show that $K_{\mathrm{DM}} = 1$, $K_{\mathrm{IPW}} = \frac{1}{2 \eta}$, and $K_{\mathrm{DR}} = \frac{\eta + 1}{\eta}$, where $\eta$ is a constant that satisfies $\mathbb{P}(\eta \leq \pi_b(X, S) \leq 1 - \eta) = 1$. Note that, in Theorem 1, we consider the ground-truth nuisance parameters because it has been shown that, under mild assumptions (e.g., uniform consistency of nuisance estimators), estimation errors in the nuisance parameters do not affect the leading term in the convergence rate (see Sec. 3.1, Sec.3.2, Lemma 4 in [1]). This is also consistent with previous work that provides generalization bounds for off-policy learning [2, 3].
> >
> >     **Action:** We moved the definition of $K_{\mathrm{DM}}$ to the main paper to clarify the dependence of the bounds on the policy score.
> >
> > ## References
> >
> > [1] Shalit et al. (2017). “Estimating individual treatment effect: generalization bounds and algorithms” ICML.
> >
> > [2] Athey and Wager (2021). “Policy learning from observational data” Econometrica.
> >
> > [3] Kallus and Zhou (2018). “Confounding robust policy-improvement” NeurIPS.
> >
> > [4] Hatt et al. (2022). “Generalizing off-policy learning under sample-selection bias” UAI.
> >
> > [5] De-Arteage et al. (2022). “Algorithmic fairness in business analytics: Directions for research and practice” Production and Operations Management.

---

### Meta-Review · Area_Chair_jtAv · 2023-12-09

**Metareview:**

This paper studies off-policy policy learning under fairness constraints. Specifically, two notions of fairness are explored. _Action fairness_ guarantees that the policy's decisions do not depend on certain "sensitive" attributes. _Value fairness_ is concerned with how different subpopulations (determined by sensitive variables) benefit from a policy's decisions, and it comes in two flavors: _envy-free fairness_ ensures that the policy's _value_ (i.e., expected utility) is more or less consistent across subpopulations, while _max-min fairness_ maximizes the worst-case performance across subpopulations. The paper proposes a two-step procedure to learn policies that satisfy both action and value fairness. A generalization bound is established, which shows that w.h.p. any policy's value and fairness constraints asymptotically (as $n \to \infty$) converge to their empirical measurements provided the policy class has low Rademacher complexity. The paper is rounded out with an empirical study on simulated and real datasets.

The reviewers agree that the paper's topic is interesting and important. They say it is a well written paper that is easy to follow. The appreciated the inclusion of generalization theory, which adds rigor to the ideas.

There were several primary critiques:
1. The algorithm depends on several "nuisance parameters," $(\mu_1, \mu_0, \pi_b)$, which must be estimated. The paper does not describe how these parameters should be estimated, or how estimation error affects the proposed method and bounds. In their response, the authors state that their focus is not on the nuisance parameters (but rather how fairness affects policy learning), so they used the ground truth nuisance parameters. They further claim that the nuisance parameters can be estimated via TARNet without affecting the algorithm's efficacy, and they ran additional experiments (discussed in the appendix of the latest revision) which they say support this claim. Reviewer `mTnw` was not convinced by this response (please refer to their review and responses for details).
2. The fairness definitions studied in the paper are not novel and have appeared (by other names) in prior work. The authors acknowledge this in their responses and state that their main contributions are not the fairness definitions.
3. The proposed method is essentially a composition of known techniques. The authors seem to acknowledge this (see their response to reviewer `TRGr `), but claim that their contributions -- the formalization and analysis of the problem, and the proposed solution -- are still valuable. I tend to side with the authors on this point, as novelty is always "in the eye of the beholder."
4. The experiments do not compare to any baselines. The authors acknowledge this and state that it is because there is no applicable prior work on fair policy learning. Nonetheless, they added an ablation study (in the appendix of the revision) that explores other approaches to the first step in their procedure.

I applaud the authors for responding to reviewer feedback and quickly incorporating it into a revision of the paper. In particular, running additional experiments on short notice is heroic. That being said, the reviewers have said (during discussion) that some of their concerns are still unresolved. (In particular, the nuisance parameters.) Also, given that there are new experimental results, I would feel more comfortable if the paper could undergo another round of review. The paper would also likely benefit from another round of polishing.

**Justification For Why Not Higher Score:**

This paper is on the border, and frankly, I would not mind it being bumped up to a poster talk. The main sticking point is that there are some unresolved concerns about how to estimate nuisance parameters, and the effect of estimated parameters, on the proposed method. The authors rushed out an experiment to try to address this concern, but the reviewers were not convinced.

**Justification For Why Not Lower Score:**

N/A

---

### Decision · Program_Chairs · 2024-01-16

Reject